# Phospholipid composition strongly affects the assembly of β barrel proteins into purified bacterial outer membranes

Thushani D. Nilaweera [1], Nathan T. Brandes [2,3], Ian S. LaCroix [2,3], Benjamin Schwarz [2] & Harris D. Bernstein [1] ✉

Virtually all integral outer membrane proteins (OMPs) produced by Gram-negative bacteria contain a unique 'β barrel' structure that serves as a membrane spanning domain. The universal barrel assembly machine (BAM) catalyzes OMP assembly (folding and membrane insertion) in vivo, and purified *Escherichia coli* BAM that is reconstituted into proteoliposomes catalyzes OMP assembly in vitro. Here we show that BAM also catalyzes the assembly of OMPs into outer membrane fractions ('native OMs') that are purified by optimized conventional methods. Interestingly, we found that OMP assembly was moderately impaired when native OMs were isolated from a *mlaA*⁻ strain that is deficient in maintaining OM lipid homeostasis but was strongly reduced when native OMs were isolated from a *pldA*⁻ strain that is deficient in a parallel pathway. We also found that the *mlaA* and *pldA* deletions altered the OM phospholipid profile to different degrees that correlated with the degree to which the mutations impaired OMP assembly. Taken together, our results provide direct evidence that the *mla* and *pldA* pathways play distinct roles in maintaining OM homeostasis and strongly suggest that OM phospholipids play a more significant role in OMP biogenesis than previously appreciated.

A variety of pathogenic bacteria have acquired resistance to currently available antibiotics and are increasingly becoming multi-drug resistant 'superbugs'[1–3]. Many of these pathogens are Gram-negative bacteria such as *Acinetobacter baumannii*, *Pseudomonas aeruginosa* and members of the Enterobacteriaceae[1–6]. The development of new antibiotics that target Gram-negative bacteria is challenging due to their distinctive cellular architecture[4]. Unlike Gram-positive bacteria, which are surrounded by one cell membrane, Gram-negative bacteria are surrounded by two cell membranes, an inner membrane (IM) and an outer membrane (OM)[7,8]. In contrast to the lipid bilayer of most biological membranes, the lipid bilayer of the OM is asymmetrical in that the inner leaflet is composed of phospholipids (PLs), while the outer leaflet is composed of a unique glycolipid called lipopolysaccharide (LPS) that prevents the uptake of hydrophobic molecules and thereby

provides the first line of defense against environmental toxins and many available antibiotics[8–10]. Furthermore, the OM is unique in that it is densely packed with a distinct type of integral membrane protein. Unlike typical integral membrane proteins that contain one or more hydrophobic α-helical transmembrane domains, almost all bacterial integral outer membrane proteins (OMPs) contain an amphipathic β sheet that folds into a cylindrical structure known as a 'β barrel' that anchors the protein to the membrane[8,10]. The first and last β strands of β barrels form a hydrogen bonded seam that confers extreme stability. OMPs mediate a variety of functions often related to molecular transport, OM biogenesis and homeostasis, and pathogenesis[11]. OMPs are highly diverse in sequence and structure, and their β barrels contain 8-36 β strands[12–14]. While some OMPs simply consist of empty β barrels, others contain a segment that is embedded within the β barrel

[1]Genetics and Biochemistry Branch, National Institute of Diabetes and Digestive and Kidney Diseases, National Institutes of Health, Bethesda, MD, USA. [2]Proteins and Chemistry Section, Research and Technologies Branch, Rocky Mountain Laboratories, National Institute of Allergy and Infectious Diseases, National Institutes of Health, Hamilton, MT, USA. [3]These authors contributed equally: Nathan T. Brandes, Ian S. LaCroix. ✉e-mail: harris_bernstein@nih.gov

pore[15]. In addition, some OMPs contain extracellular and/or peri-plasmic domains[11,13], and some OMPs form homo/hetero-oligomers[8].

During OMP biogenesis (Fig. 1, left), newly synthesized OMPs are first translocated into the periplasm through the Sec complex and then bound by molecular chaperones such as SurA, Skp and DegP that maintain them in an insertion-competent conformation[16–18]. SurA appears to be the most important chaperone in that the deletion of *surA* causes significant defects in OM integrity and a major reduction in the steady-state level of OMPs[19–21]. Subsequently, OMPs are targeted to the OM where their β barrels are assembled (folded and inserted into the OM) by the hetero-oligomeric barrel assembly machine (BAM)[22,23]. While the composition of BAM is variable[24–26], all Gram-negative bacteria produce BamA, an OMP that contains a β barrel plus multiple periplasmic polypeptide transport associated (POTRA) domains[25], and BamD, a lipoprotein. Both are essential for survival[27], although cells can live without BamD under specific conditions[28]. In *E. coli*, BAM also contains three non-essential lipoproteins (BamB, C and E)[23]. Several models have been proposed to explain the mechanism by which BAM catalyzes OMP assembly, but all of the models are based on the observation that the BamA β barrel can open laterally (and thereby distort the local lipid bilayer) because its first and last β strands are held together by only a few hydrogen bonds[29,30]. Several recent structural and biochemical studies favor a model in which the membrane insertion of OMPs is initiated by the binding of a conserved C-terminal sequence ('β signal') to the first β strand of BamA in an open conformation[31–33]. In the later steps, the new OMP enters the OM as a curved β sheet which then forms a hybrid barrel with the BamA β barrel, barrelizes by folding inward, and then closes in a strand exchange reaction[33–35]. Interestingly, recent atomic force microscopy studies revealed that OMPs and LPS are not randomly dispersed in the *E. coli* OM but are rather phase separated into OMP-rich and LPS-rich regions[36–38]. Available evidence indicates that higher order OMP assemblies known as 'OMP islands'[36] center around abundant OMPs such as OmpC and OmpF and that the OMPs are connected by a single

PL-LPS pair[37–39]. Although BAM is also located in clusters[40–42] that are randomly distributed throughout the OM (and that might facilitate OMP island formation), most OMPs appears to be inserted near the middle of the cell[36] because mature tetrapeptide-rich peptidoglycan, which is concentrated away from the cell septum, binds to BAM and blocks its function[40].

While only a few OMPs are required for viability or OM integrity, the maintenance of lipid asymmetry in the OM is critical for survival in harsh environments[43]. Besides producing a conserved and essential machine that constitutively transfers LPS from the IM directly to the outer leaflet of the OM (the Lpt complex; Fig. 1, middle)[10], many Proteobacteria remove mislocalized PLs by at least two distinct pathways: (a) the Mla pathway (MlaABCDEF), which extracts mislocalized PLs from the outer leaflet of the OM and uses the ATPase activity of MlaF to transport them back to the IM[43–45], and (b) phospholipase A (PldA) or PldA analogs (e.g., MlaYZ), which degrade mislocalized PLs and increase LPS synthesis through a regulatory cascade[46,47] (Fig. 1, right). *E. coli* and *P. aeruginosa* strains that lack both pathways are highly sensitive to hydrophobic compounds such as SDS[43,47]. Available evidence suggests that the Mla pathway also promotes anterograde PL transport, possibly through passive diffusion[48,49]. Furthermore, members of the AsmA family of proteins (e.g., TamB, YdbH and YhdP) that have been associated with anterograde transport of PLs might also play a role in maintaining OM lipid homeostasis[50,51]. Interestingly, studies conducted in the 1970s provided evidence that all of the major classes of lipids [phosphatidylethanolamine (PE), phosphatidylglycerol (PG), and cardiolipin (CL)] are present in the *E. coli* OM and are either enriched or depleted relative to the IM[52,53]. The physiological significance of these findings is unknown, however, and it is unclear if there is a more nuanced distribution of PLs based on class subtypes (e.g., the length of acyl chains) because these studies were conducted before the advent of modern advances in mass spectrometry.

Despite the complexity of the bacterial OM, OMP assembly has been replicated in vitro using purified components. It has been shown

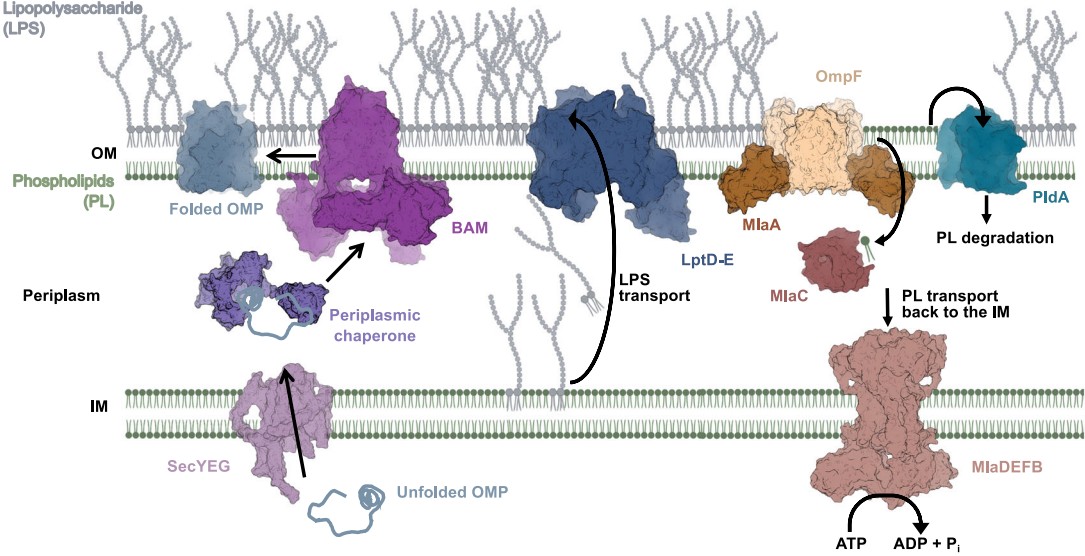

**Fig. 1 | Bacterial OM biogenesis.** After newly synthesized outer membrane proteins (OMPs) are transported across the inner membrane (IM) via the Sec machinery (only SecYEG is shown, PDB ID:6R7L[111]) they interact with periplasmic chaperones (e.g., SurA, PDB ID: 1M5Y[112]) which maintain them in an insertion-competent conformation and target them to the barrel assembly machine (BAM; PDB ID: 5D0O[113]), a hetero-oligomer that catalyzes their insertion into the outer membrane (OM) and their folding into a β barrel structure. LPS is transported by the Lpt complex from the outer leaflet of the IM directly to the outer leaflet of the OM, where it is inserted by LptD-E (PDB ID: 4Q35[114]). Mislocalized phospholipids (PLs) are removed from the outer leaflet by phospholipase PldA (PDB ID: 1QD6[115]) (or by PldA analogs) or MlaA, a lipoprotein, bound to the porins OmpF (PDB ID: 5NUP[116]) or OmpC, that facilitates their transport back to the IM via periplasmic protein MlaC (PDB ID: 5UWA[117]) and deliver them to MlaDEFB (PDB ID: 6ZY3[49]). The retrograde transport reaction requires the ATPase activity of MlaF. Members of the AsmA family of proteins are believed to transport phospholipids from the IM to the inner leaflet of the OM (not shown). For simplicity, the periplasmic and IM components of the Lpt pathway are not shown. The protein structure images were generated using Pymol Molecular Graphics System v. 2.1[110]. The lipid bilayers were created using BioRender. Nilaweera, T. (2026) https://BioRender.com/je4ofri.

that in the presence of SurA, a variety of *E. coli* OMPs that are purified from inclusion bodies and urea-denatured or synthesized de novo in a coupled transcription/translation system assemble into PL vesicles that contain purified BAM around physiological pH[54–57]. It has also been shown that OMPs synthesized in spheroplasts assemble into BAM proteoliposomes[58], urea denatured OMPs assemble into outer membrane vesicles (OMVs)[59] that are naturally shed from Gram-negative bacteria but that contain a skewed sampling of proteins[60–63], and de novo synthesized OMPs assemble very slowly into crude *E. coli* microsomal membranes (EMM) purified in the presence of EDTA[42], which strips LPS from the OM[64,65]. One drawback of all of these studies is that the membrane vesicles lack the asymmetric lipidome and full complement of resident OMPs found in an authentic OM environment. Although the lipid composition of BAM proteoliposomes only modestly affects the efficiency of assembly of urea denatured OMPs[55], the role of lipids and resident OMPs in OMP assembly under physiological conditions is unclear. Recent studies, however, have strongly suggested that interactions between LPS and OmpC facilitate its assembly[66,67]. Furthermore, OMP-lipid charge interactions drive OMP assembly into asymmetric pure lipid bilayers generated using cyclodextrin exchange[68].

In this study, we sought to determine whether BAM present in purified OM fractions that provide a more comprehensive representation of the native OM landscape than BAM proteoliposomes, OMVs, or EMM can catalyze the assembly of de novo synthesized OMPs. We first purified OM fractions (that we define as 'native OMs') from a wild-type strain of *E. coli* using classical purification methods and found that native OMs purified using an optimized sarkosyl extraction protocol promoted efficient (~40%) BAM-mediated OMP assembly. We then found that the efficiency of OMP assembly was significantly reduced when we used native OMs purified from mutant strains that are impaired in OM lipid homeostasis. An analysis of the PL profile of wild-type cells using liquid chromatography tandem mass spectrometry (LC-MS/MS) confirmed the results of earlier studies, identified many new lipids, and revealed a far more complex distribution of individual lipids than previously observed. Interestingly, we identified distinct PL profiles for the mutant strains that are potentially influenced by a feedback mechanism or crosstalk between the cell membranes and found that the degree of enrichment of CL and PG lipids in the OM along with the depletion of lyso-PLs (which might impact the membrane charge and curvature), correlated with the degree to which BAM activity was impaired. Taken together, our results provide evidence that the OM phospholipidome plays a more significant role in OMP biogenesis than previously thought and suggest that native OMs purified from different strains or even different organisms can be used in in vitro assays to identify factors that potentially affect the OMP assembly process.

## Results

### BAM located in purified native OM fractions assembles OMPs

To overcome the limitations of previously described cell-free OMP assembly assays, we first explored the possibility of using two well-established methods (sarkosyl extraction and sucrose gradient fractionation)[69,70] to purify the OM to study BAM activity in a more physiological environment. We first optimized cell growth conditions and the OM purification protocol using the wild-type *E. coli* strain MC4100 harboring a plasmid (pJH114) that expresses BAM[56]. Because divalent cations play a critical role in maintaining the integrity of the OM[8,45], the cells were grown either in LB medium (LB) or LB supplemented with $Ca^{2+}$ and $Mg^{2+}$ ('LB + MC'). After BAM expression was induced to increase the level of its activity, the OMs were purified using either sarkosyl extraction or sucrose gradient fractionation and each OM fraction was then divided in half. One half was washed with Tris pH 8.0 containing $CaCl_2$ to maintain the presence of a divalent cation ('C + ') while the other half was washed without $CaCl_2$. The four purified

OM samples [designated LB, LB(C + ), LB + MC and LB + MC(C + )] were then characterized for the presence of intact BAM and BAM orientation using trypsin digestion assays[55,71] and Western blotting with antisera against several BAM subunits. Trypsin cleaves the POTRA domains of BamA molecules that are surface exposed but does not cleave the fully integrated BamA β barrel. Based on the detection of similar amounts of BamA (Figs. S1A and S2A, lanes 1, 3, 5, 7), BamB (Figs. S1B and S2B, top blots), and BamD (Figs. S1B and S2B, bottom blots) regardless of the growth condition and the purification approach, the purified OMs retained intact BAM. Furthermore, ~90% of the full-length BamA (~90 kDa) in the purified OMs was converted to the BamA β barrel (~45 kDa) by trypsin (Figs. S1A and S2A, lanes 2, 4, 6, 8). This observation indicates that almost all of the BAM is accessible to incoming OMPs.

Next, we used a previously described OMP assembly assay[54] to evaluate the functionality of the BAM in the purified OM ('native OM') samples. In this assay, we initially used a derivative of EspP designated EspPΔ5' (previously called EspP β+46[56,72]) as a model protein. EspP is a member of the autotransporter family of OMPs that contains a large N-terminal extracellular ('passenger') domain as well as a C-terminal 12-stranded β barrel domain. EspP and derivatives that have truncated passenger domains, including EspPΔ5', which contains the last 26 residues of the passenger domain plus an N-terminal His-tag (Fig. 2A), undergo an intra-barrel autocleavage reaction following the complete translocation of the passenger domain across the OM and the folding and insertion of the β barrel domain into the OM[56,73]. EspPΔ5' was synthesized de novo using the PURExpress coupled in vitro transcription/translation system[54,74] for 30 min at 37 °C in the presence of a native OM fraction that was normalized to contain 2 μM BAM (Figs. S1C and S3). Because a fluorescent dye (BODIPY-FL) attached to lysine residues was randomly incorporated into the protein during its synthesis, assembly was assessed at the end of the incubation period by determining the fraction of fluorescently labeled EspPΔ5' (~37 kD) that was converted to the cleaved β barrel domain (~30 kD; the passenger domain fragment is too small to be detected by SDS-PAGE)[54]. A significant percentage of the EspPΔ5' was assembled in the presence of SurA and any purified OM sample obtained by sarkosyl extraction (Fig. S1D, lanes 2-5), but not in the absence of an OM sample (Fig. S1D, lane 1). A higher level of folding was detected in the presence of native OMs purified from cells grown in LB + MC than in LB, while the addition of $Ca^{2+}$ during the late stages of purification did not significantly increase assembly. Moreover, the synthesis of EspPΔ5' in the presence of OMs purified by sarkosyl extraction was much more reproducible than in the presence of OMs purified by sucrose gradient fractionation (Fig. S4), presumably because the latter OMs had a variable level of contaminants that interfered with transcription and/or translation. For these reasons, all of the native OMs that we used in subsequent experiments were purified by sarkosyl extraction using the LB + MC protocol.

After optimizing growth conditions and the OM purification protocol, we slightly modified the assembly assay in order to follow the fate of the small cohort of OMP molecules synthesized near the beginning of the transcription/translation reaction. In the modified assay the plasmid that encodes EspPΔ5', molecular chaperones and all other components except the purified native OMs were added to the reaction and incubated at 37 °C for 5 min (Fig. 2B). Subsequently, the translation re-initiation inhibitor Oncocin112 (Onc112)[75,76] was added to the reaction to prevent further synthesis of the substrate. After a 3 min incubation, purified OMs with BAM levels normalized to 1 μM were added and the reaction was incubated for up to another 30 min at 37 °C. We next performed a few additional experiments to confirm that the assembly of EspPΔ5' was mediated by BAM that was present in the native OMs. As previously reported[54], we found that SurA was required for EspPΔ5' assembly (Fig. 2C). Furthermore, as in experiments in which we examined the assembly of OMPs into BAM

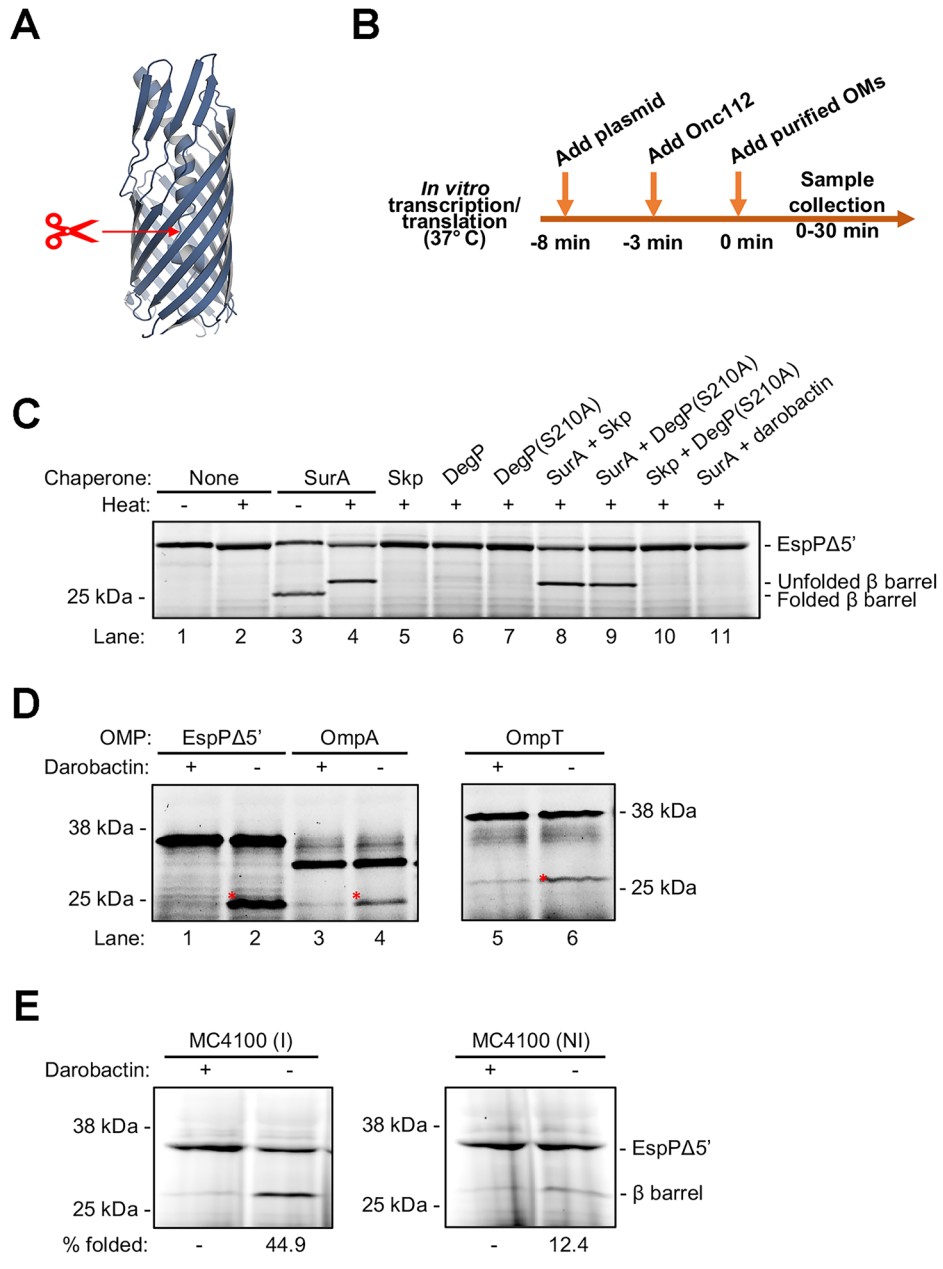

**Fig. 2 | BAM present in native OMs purified by sarkosyl extraction catalyzes OMP assembly. A** The crystal structure of the EspP β barrel and linker domains contained in EspPΔ5' (PDB ID: 3SLJ[118]). The site at which an intra-barrel autocatalytic cleavage occurs after the completion of assembly is shown. This image was created using Pymol Molecular Graphics System version 2.1[110]. **B** A schematic representation of the experimental design that we used is shown. In **C**–**E** samples were collected after a 30 min incubation at 37 °C. The fluorescently labeled OMPs were either heated (**C** and **E**) or unheated **D** and resolved by SDS-PAGE. Fluorescently labeled OMPs were visualized at an excitation wavelength of 488 nm. In the case of EspPΔ5' both the uncleaved protein and the β barrel that was generated by self-cleavage after the protein was assembled could be observed. **C** The assembly of de novo synthesized EspPΔ5' in the presence of the indicated periplasmic chaperone(s) (8 μM each) and native OMs containing 1 μM BAM that were purified from MC4100 after BAM expression was induced. **D** The assembly of de novo synthesized OMPs in the presence of 8 μM SurA and OMs purified from MC4100 in which BAM expression was induced. The OMs containing 1 μM BAM were pre-incubated with and without darobactin (10 μM). The asterisks show the fluorescently labeled folded form of each OMP β barrel. **E** The assembly of de novo synthesized EspPΔ5' in the presence of 8 μM SurA and native OMs purified from MC4100 in which BAM expression was either induced (I) or uninduced (NI). In either case, the OMs contained 1 μM BAM, and the OMs were pre-incubated with or without darobactin. The experiments shown in (**C**) and (**D**) were repeated three times and the experiment shown in (**E**) was repeated four times with similar results.

proteoliposomes[54,56], we found that a SurA to BAM ratio of 8-10:1-far less than the >3000:1 ratio that was added in OMV-dependent assays[59]—was sufficient to maximize assembly (Fig. S5). The observation that the cleaved β barrel migrated more rapidly than its predicted molecular weight in the absence of heat (and is therefore 'heat modifiable', like most properly folded OMP β barrels[77]) indicates that SurA helps to assemble EspPΔ5' correctly (Fig. 2C, lanes 3-4). Furthermore,

consistent with the results of a previous study on BAM-mediated assembly in vitro[54], EspPΔ5' assembly was detected only in reactions supplemented with SurA either alone or in combination with Skp and DegP S210A (a protease-deficient mutant of DegP) but not with Skp, DegP, or DegP S210A alone (Fig. 2C, lanes 3-10). Finally, to corroborate the conclusion that BAM catalyzes the assembly of EspPΔ5' into purified native OMs in concert with SurA, we showed that the BamA-

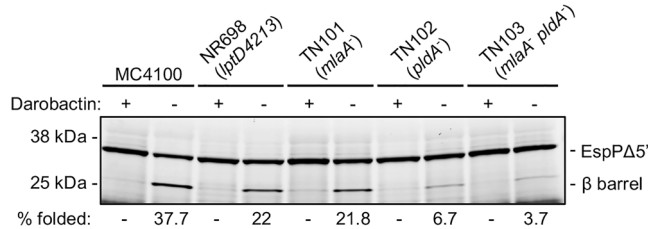

**Fig. 3 | The efficiency of EspPΔ5′ folding is reduced in the presence of native OMs purified from mutant strains deficient in OM lipid homeostasis.** The assembly of de novo synthesized EspPΔ5′ in the presence of 8 μM SurA and native OMs purified from the indicated strain after BAM expression was induced was assessed. The OMs contained 1 μM BAM and were pre-incubated with or without darobactin (10 μM). The experimental scheme shown in Fig. 2b was used and samples were collected after a 30 min incubation at 37 °C. After SDS-PAGE, the fluorescently labeled proteins were detected as in Fig. 2. The experiment was repeated twice with similar results.

specific inhibitor darobactin (which prevents the binding of β signals to the first β strand of BamA)[5,78] blocks assembly even in the presence of SurA (Fig. 2C, lane 11). Using the heat-modifiability property of OMPs, we also found that other OMPs that were synthesized de novo, including OmpA and OmpT, were inserted into native OMs in the presence of SurA (although less efficiently than EspPΔ5′) but not in the presence of darobactin (Fig. 2D). One of the proteins we tested (OmpG) folded extremely inefficiently and another protein (BtuB) did not fold at all.

## BAM mediated assembly of EspPΔ5′ into purified native OMs is affected by the OM composition

We next sought to determine whether the density of BAM in the purified OM samples affected assembly into the native OM fractions. First, we grew cells in the presence and absence of IPTG and purified the OMs. Then the folding assay was repeated using OMs purified from cells in which BAM expression was induced (I) or non-induced (NI) after normalizing the level of BAM to 1 μM. We found that the efficiency of EspPΔ5′ assembly was considerably higher when the OMs were obtained from cells in which BAM expression was induced and was similar to the level seen when de novo synthesized OMPs were assembled into proteoliposomes that contained only purified BAM[54] (Fig. 2E). This observation suggested that the density of BAM in the purified OMs rather than just the total amount of BAM added to the reaction affected its activity, and provided the first indication that the composition of the native OMs might influence the efficiency of OMP assembly.

In light of recent evidence that lipids play a role in OMP assembly[66,68], we wondered if the use of native OMs purified from strains that are defective in OM lipid homeostasis might affect the efficiency, kinetics, or mechanism of EspPΔ5′ assembly. To address this question, we next used the method described above to purify native OMs from strains NR698 (MC4100 lptD4213)[79], which has a defect in LptD biogenesis[80], TN101 (MC4100 mlaA::kan), which lacks a factor that promotes the retrograde transport of PLs that are mislocalized to the outer leaflet of the OM to the IM, TN102 (MC4100 pldA::kan), which lacks a phospholipase that degrades mislocalized PLs in the outer leaflet of the OM, and TN103 (MC4100 ΔmlaA pldA::kan), which lacks both MlaA and PldA. Similar amounts of BamA, BamB, and BamD were detected in all of the samples, and >95% of the BamA was converted to a C-terminal ~48 kD fragment that corresponds to the β barrel domain by trypsin digestion (Fig. S6). Like the native OMs purified from MC4100, the purified native OMs from the mutant strains all promoted EspPΔ5′ assembly unless darobactin was added to the reaction (Fig. 3). Interestingly, however, the efficiency of assembly was significantly reduced. While ~38% of the EspPΔ5′ was assembled in the presence of

OMs purified from MC4100 after a 30 min incubation at 37 °C, ~22% was assembled in the presence of OMs purified from NR698 and TN101, and only ~4-7% was assembled in the presence of OMs purified from TN102 and TN103. The observation that treating native OMs purified from the different strains with lysozyme to remove any residual peptidoglycan did not significantly affect EspPΔ5′ assembly (Fig. S7) strongly suggests that the OM environment itself can strongly influence BAM activity.

By scaling up our folding assay and removing samples at different timepoints as previously described[54], we found that the assembly of EspPΔ5′ was observed in the presence but not in the absence of OMs purified from MC4100 (Fig. 4A and S8A), and the level of assembly (~40% at the 30 min timepoint) was only slightly lower than the level of assembly into BAM-POPC proteoliposomes[54]. Although assembly was detected in the presence of OMs purified from NR698 (lptD4213) and TN101 (mlaA⁻) OM by the 1 min timepoint, the fraction of protein that was ultimately assembled was significantly lower than the fraction that was assembled into OMs purified from MC4100 (Fig. 4B and S8B, C). Remarkably, the free EspPΔ5′ β barrel was barely detected until the 2.5 min timepoint when OMs purified from TN102 (pldA⁻) or TN103 (mlaA⁻ pldA⁻) were added to the reaction, and in both cases only ~7.5% of the EspPΔ5′ was folded by the end of the 30 min incubation (Fig. 4B and S8D, E). Interestingly, the time required to reach 50% maximum assembly ($t_{1/2}$) for reactions that contained OMs purified from NR698, TN101 and TN103 was ~2-3 min and was not significantly longer than the $t_{1/2}$ for reactions that contained OMs purified from MC4100 (~2.4 min) (Fig. 4C). Nevertheless, assembly was notably slower when the OMs were purified from TN102 ($t_{1/2}$ ~ 3.5 min) (Fig. 4C). The kinetics data suggest that changes in the native OMs purified from TN102 affect the functionality of BAM (i.e., its rate of catalysis) while the changes in native OMs purified from NR698, TN101 and TN103 affect the fraction of BAM that can drive OMP assembly more than its functionality. It is also noteworthy that EspPΔ5′ was assembled into native OMs purified from MC4100 more slowly than BAM-POPC proteoliposomes ($t_{1/2}$ ~ 1 min)[54]. This finding suggests that assembly kinetics are also significantly affected by the major chemical and structural differences between the two types of membrane.

We next wanted to determine if the loss of BAM activity in the native OMs obtained from the mutant strains was due to a reduction in the level of OMPs or a change in the morphology and/or size of the purified OM vesicles. Consistent with previous results[43,79], we found that at log phase the levels of eight model OMPs (BamA, BtuB, LamB, FadL, FepA, OmpA, OmpC and OmpT) in vivo was similar in strains MC4100, NR698 (lptD4213), TN101 (mlaA⁻) and TN102 (pldA⁻) (Figure. S9). Although the effect of deleting both mlaA and pldA on OMP levels has not been reported, we also observed similar levels of all the OMPs in strain TN103 (mlaA⁻ pldA⁻). Furthermore, images obtained by transmission electron microscopy (TEM) after negative staining showed that the purified OM fractions used in the assays contained vesicles that were heterogenous in size and up to ~200 nm in diameter (Figure. S10). Presumably, the presence of a large number of periplasmic lipoproteins forces most of the purified OMs into an inside-out orientation that explains the trypsin sensitivity of almost all of the BamA. The vesicles in the NR698 sample seemed somewhat smaller than those in the MC4100 sample (which might be related to the hypersensitivity of NR698 to antibiotics and/or their flat, rugose colony morphology), and those observed in the TN103 sample appeared to have a double-layer morphology. The vesicles in the TN101 and TN102 samples, however, had a more normal appearance. Thus, we did not find a significant correlation between vesicle size or shape and BAM activity. Taken together, the analysis of OMP levels in vivo and the TEM images suggested that the impairment of lipid asymmetry and/or the change in the lipid composition of the native OMs purified from the mutant strains accounts for the loss of BAM activity under our experimental conditions.

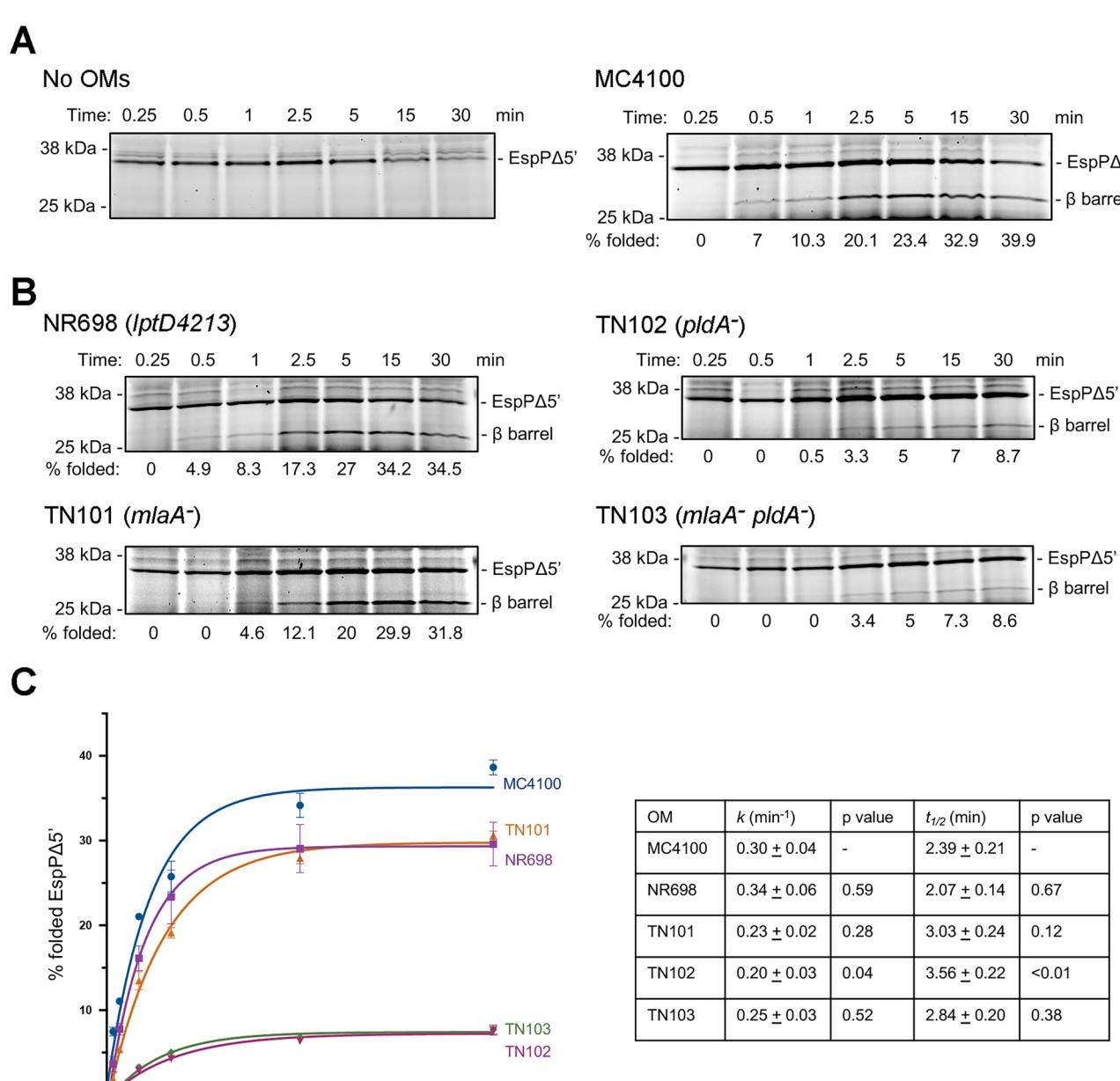

**Fig. 4 | Kinetics of EspPΔ5' assembly into native OMs purified from MC4100 and mutant strains. A** The assembly of de novo synthesized EspPΔ5' in the presence of 8 μM SurA and either the absence (left) or presence (right) of OMs purified from MC4100 after BAM expression was induced (containing 1 μM BAM) was monitored over a 30 min time course using the experimental scheme illustrated in Fig. 2b. A representative experiment is shown. **B** The experiment shown in a was repeated except the native OMs were derived from the indicated strain. A representative experiment is shown. The experiments shown in (**A**, left) was repeated three times

and (**A**, right) and (**B**) were repeated four times with similar results. **C** The percent of EspPΔ5' that was folded in the presence of SurA and OMs purified from the indicated strain over a 30 min time course is shown. The average values of four independent experiments are indicated in the plot. The error bars represent the standard error. The rate constant ($k$) and the $t_{1/2}$ were calculated from a single-exponential fit, and the values for each strain were compared to those of MC4100 via one-way ANOVA followed by a Dunnett test. $p < 0.05$ was considered statistically significant.

## Mutations that impair OM lipid homeostasis progressively alter the OM to whole cell PL differential

Because our current understanding of the *E. coli* OM phospholipidome is based on studies that were performed well before the advent of modern MS and its routine application to lipidomics[52,53,81], we developed a novel LC-MS/MS method to characterize changes in the lipid composition of the OM at the level of individual lipid molecules. The scope of this method was based on *E. coli* lipid data from previous reports[82-87] that describe the presence of the PL classes PE, PG, CL and

their associated lyso-PLs [lysophosphatidylethanolamine (LPE) and lysophosphatidylglycerol (LPG)], with each lipid class containing a range of acyl chains from 12-20 carbons in length including cyclopropane fatty acids (Ccy17 and Ccy19) (see Methods). The method leveraged conserved fragmentation behavior for PLs under negative polarity ionization which yielded diagnostic fatty acid fragment ions to measure each target lipid at both the lipid class and acyl level. To ensure maximum coverage of PLs present in any of the samples that we analyzed in this study, the method was calibrated on a test mixture of

all of the wild-type and mutant strains prior to data collection. In total, 405 individual lipids spread across the lipid classes indicated above were targeted in the final method. In our data analysis, the signal of each PL was normalized to the sum of the PL signals in that sample to offset slight differences in yields between samples.

Initially, we characterized the PL profile of an MC4100 (WT) whole cell lysate after inducing BAM expression [which we designated the whole cell (WC) phospholipidome] and native OM fractions purified from the same cultures using the sarkosyl extraction method described above. In the resulting dataset, 257 lipids were positively identified and measured. A clear enrichment was evident across nearly all PE species in the OM fraction with a fold increase approaching 2.5 for the most enriched PE species (Fig. 5A). In contrast, PG lipids were nearly universally depleted in the OM fraction with a maximum fold decrease approaching −4 compared to the WC phospholipidome. CL lipids displayed overall OM depletion but with a mixed response showing ~90% of significantly affected CLs ($n = 53$) decreasing and ~10% ($n = 6$) increasing in the OM fraction compared to the WC phospholipidome. Interestingly, lyso-PLs originating from both PE and PG were significantly enriched or trended toward enrichment in the OM. The overall increase in lyso-PLs, especially LPGs, which increase in the OM despite a general depletion of PGs, suggested a concentration of general phospholipase activity in the OM, possibly associated with PldA. Although much more detailed, our analysis of the OM phospholipidome is remarkably consistent with previous results obtained using thin layer chromatography and other older methods[52,53,81].

To determine if the mutations in the strains that impair OMP assembly in our in vitro assay affect the OM phospholipidome, we next compared the lipid profiles of the WC and purified native OM samples from each mutant strain. The global variance in the lipidome between WT and mutant strains and the lipid differential of the OM were examined by principal component analysis (PCA) (Fig. 5B). The vector encapsulating the largest percent variance in the dataset characterized distinction between the WC and the OM phospholipidome (PC1), and the second largest variance vector (PC2) described strain-specific changes that were conserved between the WC and OM samples from each strain. As we expected given the results of the OM vs. WC comparison in the WT strain (Fig. 5A), PC1 was driven by a divergence of the PE and PG variance in the dataset, with PE being positively weighted and PG being negatively weighted, while PC2 was driven by intraclass variance in both PE and PG (Fig. 5C). In a control experiment we found that the levels of two highly abundant OMPs (OmpC and OmpA) were very similar in all of the samples we used for our lipidomic analysis while there was slightly more variability in the levels of less abundant OMPs in the OM samples that appear to be due primarily to experimental error associated with sample preparation (Fig. S11). Consistent with previous work on various Lpt mutant strains[88,89], we also found that the levels of LPS were similar in all of the WT and mutant WC samples (Fig. S11), even in samples derived from strain NR698 (*lptD*4213). Thus, the strain-specific changes in the PL profile that we observed in NR698 were most likely due to the phase-dependent changes in LptD levels that we and other investigators have observed[79,90] (Fig. S12), which in turn affected the activity of MlaA and/or PldA. The phospholipidome changes observed for strains TN101 (*mlaA*⁻), TN102 (*pldA*⁻), and TN103 (*mlaA*⁻ *pldA*⁻) were most likely due to the loss of MlaA and/or PldA activity.

Interestingly, PC1 of the initial lipidomic PCA showed a progressive loss of the WT differential between OM and WC PLs in the mutant samples, with TN103 showing the lowest OM to WC variance (Fig. 5B). To investigate these changes in the PL differential directly, we calculated the fold change of each OM sample versus the average of its cognate WC samples to isolate the OM PL differential and avoid consideration of inter-strain variance. The resulting fold change ratios were transformed to the base two logarithm (log2FC). The log2FC OM dataset was analyzed by one-way ANOVA, and lipids that passed a 10%

false discovery rate (FDR10%) filter were arranged by lipid class (Fig. 5D and S13). While there was a noticeable change in the OM lipid differential in NR698 and TN101, the overall OM lipid enrichment/depletion profiles were similar to the WT profile (Fig. S13). However, we observed a much greater reduction in the OM PL differential in TN102 and TN103 (Fig. 5D and S13). This reduction could be observed to an equal extent in OM-enriched PE lipids and OM-depleted PG lipids, but was most striking in CL lipids for which we observed a near complete loss of differential and a potential inversion. Curiously, the OM lipid differential was reversed in both TN102 and TN103 for LPG but not LPE species (Fig. 5D and S13). Taken together, the magnitude of the change in the lipid differential and the inversion of the OM differential of LPGs and CLs in TN102 and TN103 suggests that the levels of both of these lipid families in the OM are maintained through a feedback mechanism and/or crosstalk between the IM and OM.

## Mutations that impair OM lipid homeostasis drive global shifts in PL acyl chain length

In addition to showing the loss of OM lipid differentials along PC1 in the mutant strains, the initial lipidomics global PCA revealed that the mutant strains exhibited varying levels of compensation in their phospolipidomes along PC2 that were shared between the purified native OM and WC samples (Fig. 5B). To investigate this compensatory behavior, the WC and OM PL profiles were analyzed separately across all strains based on the lipid class and the specific acyl chain (Fig. 6 and S14). Strain-dependent patterns were largely conserved across lipid classes between purified OM fractions and WC samples (Fig. 6 and S14C, D). The mutant strains exhibited deviations from the WT strain distributed across all of the lipid classes. With respect to strain NR698 (*lptD*4213), the most notable deviations in both OM and WC fractions were intraclass shifts in PE lipids and reduced levels of all LPEs and LPGs. This finding was surprising because NR698 produces PldA, which degrades PE and PG that are mislocalized to the outer leaflet of the OM to generate LPE and LPG in an acyl chain length-specific (but not lipid class-specific) manner[91]. Because PldA is the only known phospholipase in the *E. coli* OM, changes in the OM induced by the *lptD*4213 mutation might have impaired its activity. As expected, the levels of lyso-PLs in both the OM and WC samples obtained from strain TN101 (*mlaA*⁻) were similar to those obtained from WT cells. The purified OM samples from TN101 and NR698, however, showed a series of similar deviations from the WT profile (primarily PE shifts) that did not clearly correlate with acyl chain length.

The absence of PldA in strains TN102 (*pldA*⁻) and TN103 (*mlaA*⁻ *pldA*⁻) resulted in the most significant changes in the OM and WC PL profiles, and the lack of both MlaA and PldA in TN103 impacted several classes of PLs synergistically. Most notably, the absence of both proteins resulted in a higher enrichment of longer chain PEs, PGs and CLs than the absence of either protein alone. Consistent with the strain-specific behavior observed in the initial PCA analysis (Fig. 5B), the OM PL profiles of TN102 and TN103 were highly distinct with intraclass differences that tracked with the length of the acyl chain (Fig. 6 and S14C, D). TN102 maintained an acyl chain pattern that more closely resembled the WT pattern while longer acyl chains were enriched in TN103 in both OM (Fig. S14A, B) and WC (Fig. S14E, F) samples. Consistent with the directionally similar but lower intensity shift of TN101 away from WT and toward TN103 along PC2 (Fig. 5B), TN101 showed a similar but much weaker pattern of longer acyl chain enrichment in both OM and WC samples than TN103 and a similar pattern of short acyl chain depletion. As expected based on the known activity of PldA[91], lyso-PLs were highly depleted in both TN102 and TN103. The contrast between the similar OM-to-WC differential patterns observed in TN102 and TN103 and the highly divergent OM and WC PL profiles in the two strains is very striking. The results suggest that while the loss of PldA has an outsized effect on the PL differential, the loss of MlaA in TN101 leads to a separate set of changes in acyl

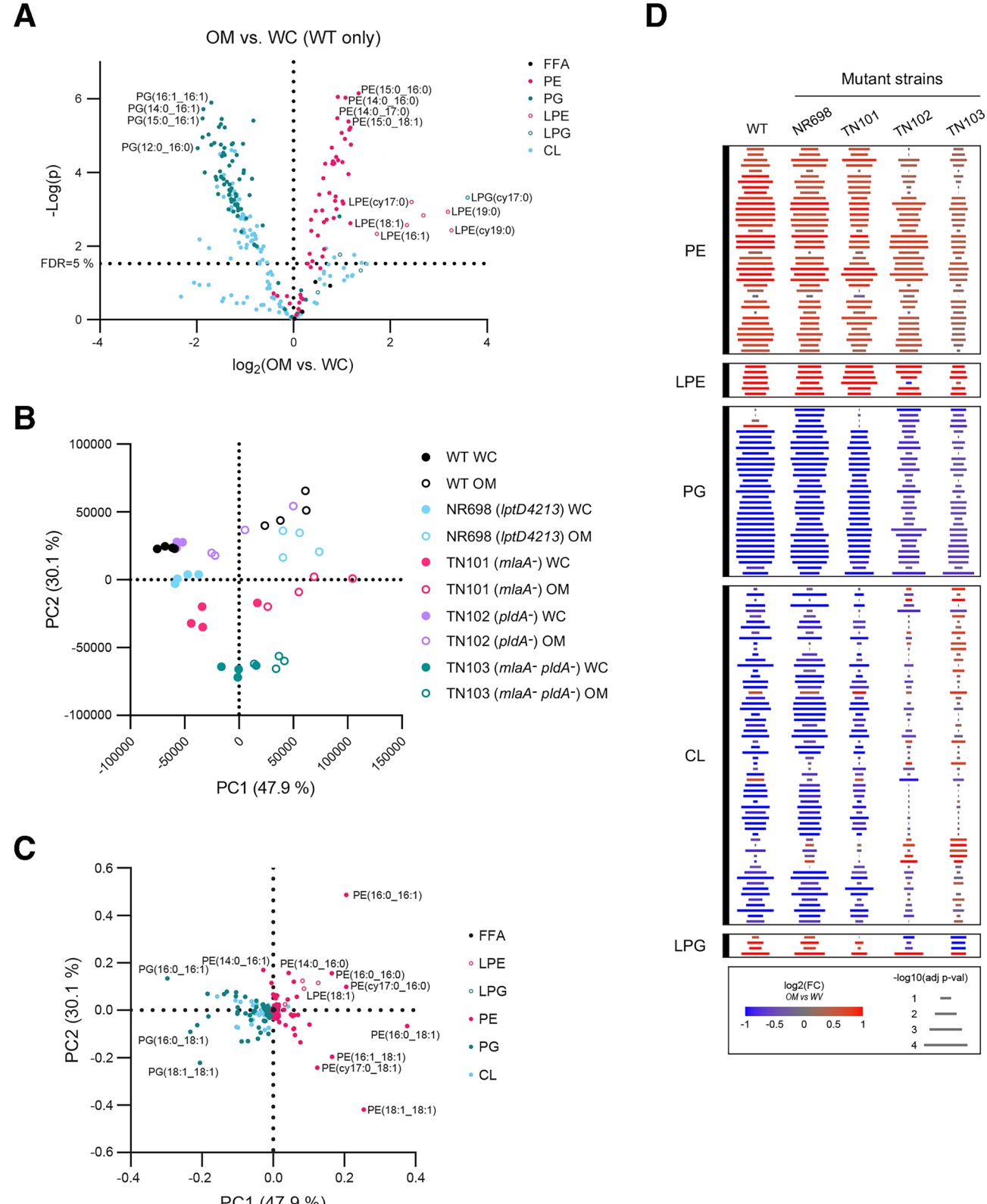

chain length in both the OM and WC PL profiles that is exacerbated by the loss of PldA in TN103. The synergistic (but non-additive) effect of the two mutations in TN103 on both the OM to WC differential and the OM (or WC) inter-strain acyl chain length comparisons suggests that PldA and MlaA-driven lipidomic processes interact strongly and are influenced by a feedback mechanism or crosstalk between the IM and OM.

## Discussion

In this study we demonstrate that de novo synthesized OMPs can be efficiently assembled into native OMs purified from *E. coli*. Although the primary model OMP used in our experiments, EspPΔ5', could be assembled into OMs purified by either sarkosyl extraction or sucrose gradient fractionation, we found that the sarkosyl extraction protocol was more reliable, presumably because sucrose gradient fractionation

**Fig. 5 | Mutations that impair OM lipid homeostasis change the OM to WC PL differential. A** Comparison of the OM to whole cell (WC) phospholipid (PL) profile fractions in WT cells by two-tailed *t*-test with a correction for multiple comparisons using the Benjamini Hochberg method reflected at a false discovery rate (FDR) of 5% (*p* = 0.032), with a lower FDR selected here due to the large magnitude of the effect. Lipids are colored according to head group class: PE phosphatidylethanolamine (closed pink circle), PG phosphatidylglycerol (closed teal circle), CL cardiolipin (closed light blue circle), LPE lyso-phosphatidylethanolamine (open pink circle), LPG lyso-phosphatidylglycerol (open teal circle), FFA free fatty acid (closed black circle). **B** Principal component analysis (PCA) on Pareto-scaled lipidomic data

from WC and OM PL datasets for WT and mutant strains. **C** Corresponding lipid loading plot for the principal components displayed in B with lipids colored by head group class as in (A). **D** Difference of difference analysis of the OM PL signature across strains. The average intrastrain fold change of each lipid in the OM vs. WC fractions was calculated and these values were compared between strains via one-way ANOVA. Lipids that passed a 10% FDR correction for multiple comparisons using the Benjamini-Hochberg method are displayed. The color of each bar reflects the base 2 logarithm of the OM to WC fold change for that lipid in that strain, and the length of the bar reflects the negative base 10 logarithm of the OM vs. WC two-tailed *t*-test for that lipid in that strain.

did not consistently remove contaminants that interfere with transcription or de novo protein synthesis. Several lines of evidence indicated that, not unexpectedly, OMP assembly was catalyzed by BAM and thereby confirmed that BAM functionality is retained in the purified OM fractions. In addition to BAM, we found that SurA was required for OMP assembly into native OMs and was the only chaperone that was effective. While similar results were reported when de novo synthesized OMPs were assembled into BAM proteoliposomes, it was shown that DegP [and DegP(S210A)] can substitute for SurA to promote the assembly of urea denatured OMPs while Skp inhibit assembly[54]. Our results confirm that the method by which model OMPs are produced dictates the functionality of periplasmic chaperones in in vitro assembly assays. Furthermore, we found that the activity of BAM in OMs purified from cells in which BAM expression was induced was considerably higher than the activity of the same amount of BAM in OMs purified from cells in which BAM expression was non-induced. This observation implies that the density, rather than the total amount of BAM in the OMs is the major factor that influences its activity. In light of evidence that BAM forms 'precincts' that contain multiple copies of the complex that swell when OMP assembly is at its highest level[40–42], it is possible that in our system overexpressed BAM forms precincts more readily and is therefore more active. Alternatively, other biochemical and/or structural features of the OM that potentially enhance BAM activity, such as the reported formation of 'OMP islands'[36,37], were preserved or replicated to a certain degree in the OMs purified from cells in which BAM expression was induced.

Interestingly, we found that mutations that impair OM PL homeostasis significantly reduced the efficiency of OMP assembly in our in vitro assay and, in the case of the *pldA* deletion, also slowed the rate of assembly. This finding suggests that changes in the native OMs from the mutant strains reduced the number of active BAM complexes while those from TN102 (*pldA*⁻) affected the functionality of individual BAM complexes as well. Given that the level of OMPs and LPS in each strain was similar, the reduction in OMP assembly was most likely due to significant changes in the PL composition of the native OMs purified from the mutant strains. In pursuit of potential PL contributions to the loss of BAM activity, we developed an *E. coli*-tailored lipidomics method that has an unprecedented depth of coverage. Consistent with the results of experiments that were conducted 50 years ago[52,53,81], our method showed that the three major PL classes that are found in wild-type *E. coli* are enriched in either the OM (PE) or the IM (PG and CL). The levels of enrichment were also similar to those observed in earlier studies (see Methods). We did not observe any bias in the acyl chain content of enriched or depleted PLs in the OM despite a previous report of a difference between the OM and the IM in unsaturated and cyclopropane acyls[53]. If an acyl chain bias exists between the IM and OM, it may not be clearly resolvable amid the large magnitude of the OM PL class differences we observed. Importantly, our results emphasize that PLs are not simply a functionless, uniformly distributed component of bacterial membranes, but rather a highly differentiated group of molecules that can shape the properties of each membrane and/or serve distinct roles in cell physiology. In this regard, the high enrichment of all forms of LPE and LPG in the OM that we observed is

especially notable because lyso-PLs have been associated with positive membrane curvature[92] and membrane remodeling[93], properties that would likely be important in the OM because membrane distortion appears to be a key element of BAM function[33]. The strong proclivity of zwitterionic PG to form lipid bilayers and negatively charged PE to form hexagonal II and cubic phase structures[93,94] might also impact the ability of BAM to catalyze OMP assembly. Finally, optimal BAM activity might require a specific lateral pressure profile, which can be affected by the lipid composition of the OM[94,95].

We also found that the OM to WC PL differential and the individual OM and WC PL profiles of native OMs purified from the mutant strains diverged from those of MC4100 to varying degrees. There were significant differences in the OM PL profile of NR698 (*lptD4213*), including a reduction in LPE and LPG levels. The results are very surprising because NR698 does not contain defects in any of the PL transport systems, and the *lptD* mutation did not significantly affect the steady-state level of LPS. It seems likely that differences in *lptD* expression at specific stages of cell growth (see Fig. S12) transiently altered LPS levels, which in turn affected PldA activity or generated a signal that affected PL trafficking and/or production. The concomitant loss of LPE and LPG might explain the reduced assembly of EspPΔ5' that we observed in our in vitro assay.

More significantly, we found that the OM PL profiles of *mlaA*⁻ and *pldA*⁻ strains not only differed considerably from that of wild-type *E. coli* but also differed considerably from each other. This result was highly unexpected because PldA and the Mla pathway are thought to be functionally redundant systems that evolved specifically to prevent the accumulation of PLs in the outer leaflet of the OM, based in part on the finding that *pldA* and *mla* mutations are synthetically lethal in SDS/EDTA and that PldA overexpression suppresses OM defects associated with *mla* mutations[43]. New evidence that the Mla pathway promotes the anterograde as well as the retrograde transport of PLs[48,49,96], however, raises the possibility that while the two systems overlap functionally, they also have distinct properties. If the Mla pathway is involved in the bidirectional flow of PLs, *mla* mutations might have a more neutral effect on the OM PL profile than *pldA* mutations, which only alter the OM phospholipidome. Indeed our observation that a *pldA* deletion had a much greater impact on the OM PL profile, the OM to WC PL differential, and the assembly of EspPΔ5' than an *mlaA* deletion is consistent with this scenario. It is also surprising that the assembly of EspPΔ5' into OMs obtained from strains that lack either PldA or both MlaA and PldA was similarly impaired even though the latter strain cannot remove mislocalized lipids and exhibited a PL profile that deviated the most from the MC4100 profile. In this regard, it should be noted that native OMs from both strains effectively lack both PldA and Mla activity because the periplasmic and IM components of the Mla pathway are absent from the OMs obtained from the *pldA* strain. In principle, only the PldA contained in the OMs derived from NR698 (*lptD4213*) and TN101 (*mlaA*⁻) can continue to degrade mislocalized PLs during the OMP assembly assays. Furthermore, BAM activity in OMs obtained from both TN102 (*pldA*⁻) and TN103 (*mlaA*⁻ *pldA*⁻) might be strongly affected by the dramatic depletion of LPE and LPG and the enrichment of PG and CL that we found was associated

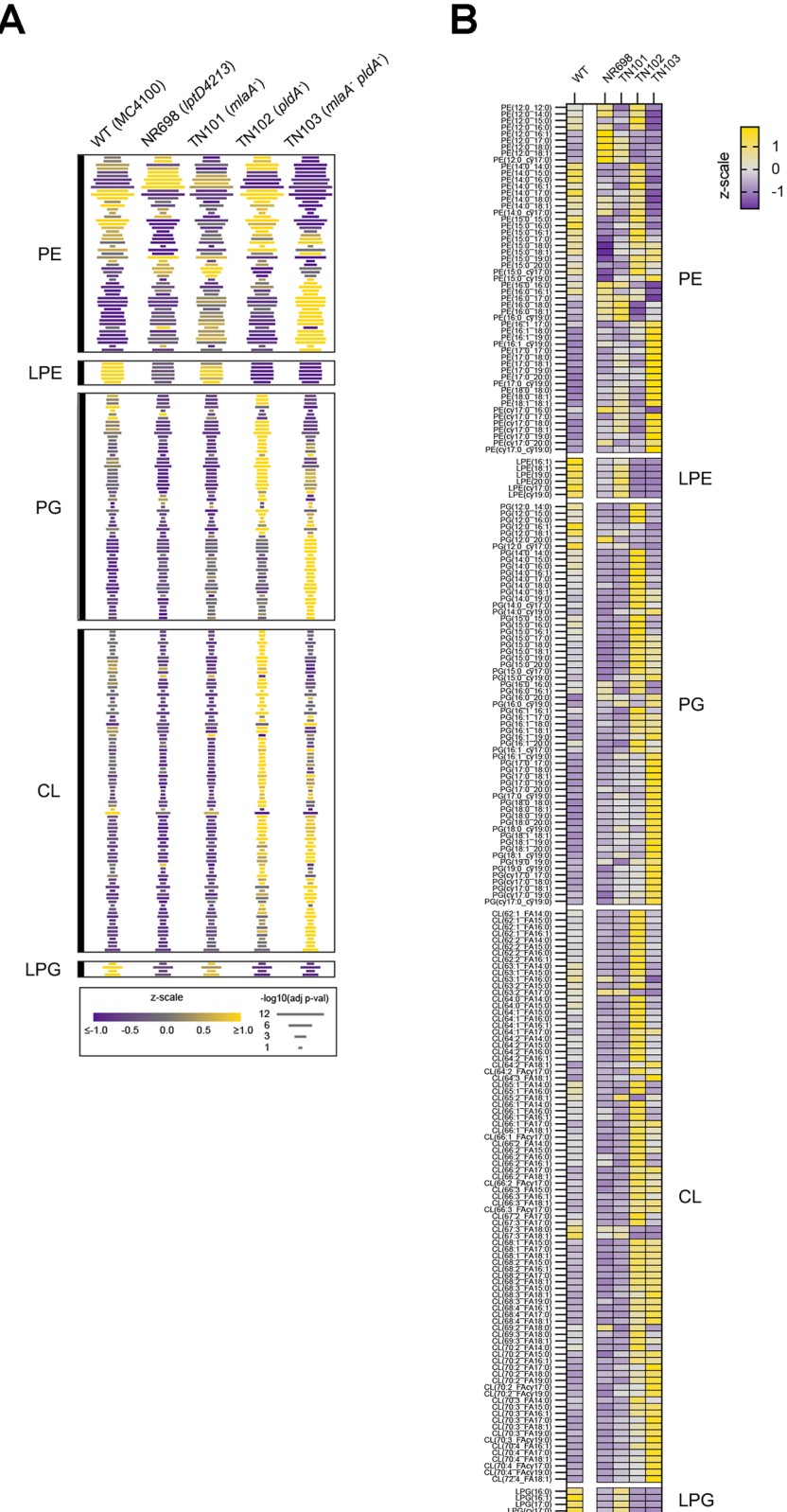

**Fig. 6 | Mutations that impair OM lipid homeostasis alter the PL profiles for purified native OMs. A** Comparison of the OM phospholipidome across strains by one-way ANOVA. Lipids that passed a filter for multiple comparisons using the Benjamini Hochberg method set at an FDR of 10% are displayed. The color of each bar reflects the z-scaled abundance within the OM dataset and the length of each row of bars reflects the negative base 10 logarithm of the ANOVA *p*-value and is thus shared across all strains for a specific lipid. Lipids are organized first according to lipid class and then according to ascending acyl chain length within each lipid class. **B** The same analysis displayed in A is shown but was expanded to display individual lipids.

with the absence of PldA. In any case, it seems likely that a change in the PL profile of this magnitude would involve feedback or intermembrane crosstalk. One caveat is that we cannot assess the contribution of PagP, which transfers palmitoyl groups from C16:0 PLs located in the outer leaflet of the OM to the lipid A moiety of LPS[97,98], to the changes that we observed in the OM PL profiles. In this regard, it is notable that while C16:0 PE and PGs appear to deviate from broader PL class-based patterns in both OM and WC samples, this deviation was shared with certain C14:0 PLs (Fig. 6, S14).

Perhaps not surprisingly, the native OM system that we developed has very different properties from lipid vesicles that contain only purified BAM. We previously found that the lipid composition of BAM proteoliposomes does not have a major impact on the folding efficiency of either EspPΔ5' or OmpA[55]. Unlike purified native OMs that contain OMPs and authentic lipids, however, BAM proteoliposomes are synthetic symmetrical PL vesicles that lack spatial organization. Perhaps because the lateral mobility of OMPs is hindered by interactions between OMPs and LPS in asymmetrical membranes but is enhanced in symmetrical PL membranes (as indicated by recent MD simulations)[37], we also found that EspPΔ5' is assembled more slowly into native OMs than BAM-POPC proteoliposomes[54]. Thus, the simple composition of BAM proteoliposomes potentially facilitates the diffusion of folded OMPs away from BAM and enhances the continuity and speed of OMP assembly. In contrast, the relatively slow assembly of EspPΔ5' into purified OMs that we observed might be due to a synergistic effect of a densely packed, static membrane environment and the unique asymmetrical architecture of the native OM.

Finally, it is striking that OMP assembly in vivo is largely unaffected by mutations that lead to the mislocalization of lipids[43,79]. The disparity between our results and the results of experiments conducted in vivo implies that although assays performed with native OMs and periplasmic chaperones are more physiological than experiments conducted with BAM proteoliposomes, they do not completely replicate events that occur in living E. coli grown under standard laboratory conditions. In contrast to the OM in growing cells, the purified OM fractions are relatively static and cannot expand in volume or promote the movement of OMPs towards the poles. Furthermore, factors located in the periplasm and IM (in addition to components of the Mla system) that are missing from our system might also enable BAM to compensate for alterations in the OM lipid bilayer. Optimal BAM function in vivo might depend on numerous synchronized and spatially coordinated processes, including the formation of phase-separated supramolecular OMP islands, membrane tension and curvature, and lipid trafficking. Nevertheless, our results might reveal the effect of significant changes in the OM PL profile on OMP assembly that occur in vivo under specific growth or stress conditions that have yet to be identified. In any case, despite the potential limitations of our assay, we show here that purified OMs can serve as an in vitro platform for studying OMP biogenesis that better mimics the native E. coli OM environment than any other platform described to date. In principle, this platform can be used to identify conditions and molecules other than PLs that affect BAM activity in vivo and to obtain further insights into BAM function.

## Methods
### Strains, growth conditions, plasmids, and antisera
The E. coli K-12 strains used in this study were MC4100 [araD139 Δ(argF-lac)169 l-e14-flhD5301 Δ(fruK-yeiR)725(fruA25) relA1 rpsL150 rbsR22 Δ(fimB-fimE)632(::IS1) deoC1], NR698 (MC4100 lptD4213)[79], TN101 (MC4100 mlaA::kan) and TN102 (MC4100 pldA::kan). TN101 and TN102 were generated by P1 transduction using strains JW2343 and JW3794 from the Keio collection[99], respectively. To construct TN103 (MC4100 ΔmlaA pldA::kan), a clean deletion of mlaA was first generated by removing the kanamycin cassette from TN101 using the Flp recombinase-producing plasmid pCP20[100] and then introducing

pldA::kan by P1 transduction. The genotype of strains TN101-TN103 were validated by PCR using the oligonucleotides listed in Supplementary Data 1. Cells were grown in LB at 37 °C at 250 rpm. Ampicillin (100 μg/ml) was added to culture media as needed, except a lower concentration (25 μg/ml) was added to NR698 cultures due to their sensitivity to the antibiotic (N. Ruiz, personal communication). The plasmids pJH114, pSK257, pET28b::espPΔ5', pET28b::ompA and pET28-b::ompT have all been described previously[55–57]. Rabbit polyclonal antisera were generated against a peptide derived from BtuB extracellular loop 2 (NH2-CDVVAYGNTGTQAQTDND-COOH), an N-terminal peptide of FepA (NH2-AEQNLQAP GVSTITADEIRKC-COOH), a C-terminal peptide of LamB (NH2-TGNADNNANFGKAVPAD FNGGC-COOH), a C-terminal peptide of BamB (NH2-CDGKLLIQAKDGTVYSITR-COOH), and full-length OmpT (purified from a gel slice). Rabbit polyclonal antisera directed against BamA, BamD, LptD, OmpA, OmpC, FadL and L4 peptides have been described previously[55,72,90,101–103]. A mouse monoclonal antibody (mAB) that recognizes E. coli LPS was purchased from Hycult Biotech (catalog number HM6011). In general, antiera were used at a 1:5000-1:10000 dilution.

### BAM expression
MC4100 transformed with pJH114 (pTRC99a-BamABCDEHis8) were grown in LB/ampicillin ('LB') or LB/ampicillin supplemented with 1 mM MgCl2 and 0.1 mM CaCl2 ('LB + MC'). Overnight cultures were diluted 1:100 into fresh medium, and cultures were grown to OD600 ~ 0.8, at which point the expression of BAM was induced for 45 min by adding 0.4 mM IPTG. In some experiments, a subset of the cells was grown in LB + MC without IPTG for an additional 45 min. NR698, TN101, TN102 and TN103 transformed with pJH114 were grown in LB + MC and BAM was expressed for 45 min as described above. At the end of the 45 min incubation, the cells were pelleted at 4000 x g, 15 min, 4 °C and stored at −80 °C.

### OM purification via the sarkosyl extraction method
The cell pellets obtained from 150 mL LB and LB+MC cultures were each resuspended in 3 mL Tris-buffered saline (TBS: 20 mM Tris, pH 7.4, and 150 mM NaCl) supplemented with 0.025 U Benzonase Nuclease (Santa Cruz Biotechnology, Santa Cruz, CA) and 40 μM 4-(2-aminoethyl) benzenesulfonyl fluoride hydrochloride. The cells were lysed by sonication on ice using a 3 mm tip (Cole-Parmer, model EW 04712-12) at 80% amplitude (30 sec on, 1 min off, 30 sec on). Cell debris was removed by centrifugation at 6000 x g, 15 min, 4 °C. The resulting supernatant was centrifuged at 100,000 x g, 1 h, 4 °C using a Beckman TLA100.4 rotor. OM fractions were purified using a modified version of a previously described sarkosyl extraction method[104]. The membrane pellet was resuspended in 375 μL 10 mM Tris, pH 8.0, containing 1% sarkosyl (Sigma, catalog number L9150), dispersed using a Dounce homogenizer, and incubated at room temperature with mixing for 30 min. At the end of the incubation period samples were centrifuged again at 100,000 x g, 1 h, 4 °C. The resulting OM pellets were resuspended and dispersed as described above in 20 mM Tris, pH 8.0, containing 0.6 mM PMSF and either 0.6 mM CaCl2 (C+) or no CaCl2, incubated for 5 min at room temperature, and then centrifuged at 100,000 x g for 1 h at 4 °C. The resulting pellets were then resuspended and dispersed again as described above in 20 mM Tris buffer, pH 8.0 and centrifuged at 100,000 x g, 1 h, 4 °C. The final pellets were resuspended in 50–100 μL 20 mM Tris, pH 8.0, flash frozen and stored at −80 °C. The native OM samples were designated LB, LB(C + ), LB + MC and LB + MC(C + ).

### Purification of native OMs by sucrose gradient fractionation
Cell pellets obtained from 600 mL MC4100 cultures grown in LB and LB + MC were processed as in the sarkosyl extraction method described above through the first ultracentrifugation step. The membrane pellets were subsequently resuspended and dispersed in 400 μL

10 mM Tris, pH 8.0, containing 20% sucrose using a Dounce homogenizer and the OM fraction was isolated using a modified version of a previously described single-step discontinuous sucrose gradient procedure[105,106]. The sucrose solutions were all prepared in 10 mM Tris, pH 8.0. The discontinuous sucrose density gradients 70%-60%-20% (70% sucrose gradient) and 73%-53%-20% (73% sucrose gradient) were prepared in 4.7 mL polypropylene centrifuge tubes (Beckman Coulter) by carefully adding each sucrose layer in the given order. For the 70%-60%-20% density gradient, 1.5 mL 70%, 1.5 mL 60% and 1.2 mL 20% sucrose were added, and for the 73%-53%-20% density gradient, 1.1 mL 73%, 2.2 mL 53% and 0.8 mL 20% sucrose were added. Half (200 μL) of each membrane pellet obtained from cells grown in LB and LB + MC was separately placed on the top of the individual tubes of 70% and 73% density gradients and the remaining tube space was filled with 20% sucrose. The samples were centrifuged at 195,000 x $g$, 4 h, 4 °C using a Beckman TLA100.4 rotor. The tubes were carefully punctured from the bottom and fractions were collected dropwise[105]. The fractions of white OM layer that contained native OMs were further processed as described above in (sarkosyl extraction method) by resuspending the native OMs first in 20 mM Tris pH 8.0, 0.6 mM PMSF with and without 0.6 mM $Ca^{2+}$ and centrifuged. Then the pellets were resuspended in 20 mM Tris, pH 8.0, and centrifuged. Finally, the native OMs were resuspended in 50–100 μL 20 mM Tris, pH 8.0, flash frozen and stored at −80° C. The samples were designated LB, LB(C +), LB + MC and LB + MC(C +) as described above.

## Characterization of purified native OMs

The concentration of BAM and the orientation of BAM subunits were analyzed essentially as described previously[55,71], except that proteins were detected by Western blotting instead of gel staining. In brief, the concentration of BAM in the native OMs was determined using 20 μM BAM purified in n-dodecyl-β-D-maltoside detergent (DDM, Anatrace, catalog number D310) as BAM-DDM standard[55,103]. After subjecting the native OMs to SDS-PAGE on 8-16% Tris-glycine mini-gels (Thermo-Fisher, catalog number XP08162BOX), the concentration of BAM was determined by Western blotting using a rabbit polyclonal antiserum generated against BamA and comparing the signal to the standard signal using ImageJ software. To validate this approach, five different concentrations of BAM were subjected to SDS-PAGE, and the BamA signals observed on a Western blot were plotted to generate a standard curve that was used to recalculate the concentration of BAM in the native OMs (Fig. S3). As expected, the BAM concentrations in the native OMs that were determined by both methods were nearly identical. The accessibility of BAM subunits was determined by incubating purified OMs in 20 mM Tris, pH 8.0 with or without 2.0 mg/mL trypsin (Sigma, catalog number T1426) for 45 min at 37 °C. At the end of the reaction, proteins were resolved by SDS-PAGE and the orientation of BAM was determined by Western blotting using the anti-BamA antiserum. The presence of the BAM lipoproteins BamB and BamD in the OMs was detected by Western blotting using rabbit polyclonal antisera generated against the two proteins.

## OMP assembly assays using purified native OMs

OMP assembly in vitro was assessed using a modified version of an assay in which OMPs are produced de novo in the PURExpress® coupled transcription/translation system (New England Biolabs, catalog number E6800)[54]. During the initial optimization of OM purification using MC4100 cells, purified OM aliquots were pre-sonicated and added to individual reactions. A typical 7.5 μL reaction contained 2 μL solution A and 1.5 μL solution B from the PURExpress system, 8 U murine RNase Inhibitor (New England Biolabs, catalog number M0314), 0.2 μL FluoroTect Green$_{Lys}$ (Promega, catalog number L5001), purified OMs (containing 2 μM BAM) or 20 mM Tris pH 8.0 as a control, and 2 μM SurA purified as described previously[57]. In brief, BL21(DE3)

(ThermoFisher Scientific, catalog number EC0114) transformed with pSK257 were grown to $OD_{600}$ = 1.0 at 37 °C and shifted to 16 °C. SurA expression was induced by adding 0.1 mM IPTG and incubating the culture overnight. The cells were harvested and resuspended in 20 mM Tris pH 8.0 buffer and lysed using a continuous flow cell disruptor (Constant Systems BT40) at 30,000 psi. Cell debris was removed by centrifugation at 6000 x g, 20 min, 4 °C. SurA was purified from the resulting supernatant by Ni-NTA affinity chromatography. The purified SurA was concentrated and dialyzed three times with 20 mM Tris buffer pH 8.0. The reactions were incubated at 37 °C for 5 min prior to the addition of pET28b::espPΔ5' (5 ng/μL) and then incubated for an additional 30 min at 37 °C with mixing at 600 rpm.

Subsequently, the above assay was slightly modified to halt the continuous reinitiation of protein synthesis and to monitor the assembly of the small population of OMP molecules synthesized at the start of the reaction. Unless otherwise indicated, native OMs purified from cells that were grown in LB + MC and harvested after BAM expression was induced were added to the reactions. At the start of the transcription/translation reactions 5 ng/μL pET28b::espPΔ5' (or pET28b::ompA or pET28b::ompT) was added to a 7.5 μL reaction mixture containing 2 μL and 1.5 μL PURExpress solutions A and B, respectively, murine RNase Inhibitor (8 U), 0.25 μL FluoroTect Green$_{Lys}$ and the indicated levels of SurA (0-10 μM). After the samples were incubated at 37 °C (with mixing at 600 rpm) for 5 min, 10 μM Onc112[75,76] was added for 3 min, and then pre-sonicated OMs (containing 1 μM BAM) or 20 mM Tris buffer pH 8.0 (as a control) was added to the reactions and incubated for another 30 min. In some experiments, the pre-sonicated OMs were incubated with or without 10 μM darobactin for 5 min at 37 °C. To optimize assembly, 8 μM SurA (an 8:1 SurA:BAM ratio) was used in all of the single timepoint and kinetics experiments. In some experiments, the same concentration of other periplasmic chaperones, including Skp (MyBioSource, catalog number MBS203478, lot 1061801), DegP, and DegP(S210A)[54] either singly or in combination, was added. To analyze assembly kinetics, the reactions supplemented with 8 μM SurA were scaled up to 60 μL and aliquots (7.5 μL) were withdrawn at specific time points (0-30 min) and placed on ice. In some experiments, the pre-sonicated OMs were incubated with and without 1 mg/mL lysozyme (Roche, catalog number 10837059001) for 10 min on ice.

## Analysis of OMP assembly

At the end of each reaction, RNaseA (0.5 mg/mL) and 2 x SDS-PAGE loading buffer (Quality Biological, catalog number 351-082-661) were added to each sample, and the samples were either heated at 98 °C for 10 min or left unheated. Proteins were resolved by SDS-PAGE on 8-16% Tris-glycine mini-gels and de novo synthesized OMPs were visualized using a Fuji FLA-9000 imager or an Amersham Typhoon scanner at the excitation wavelength of 488 nm. The percent of de novo synthesized OMP molecules that were assembled in each reaction was quantitated using ImageJ software (version 2.14.0) using the equation % folded=100 x (signal from folded protein/signal from both folded and unfolded protein). The data were fitted to exponential decay curves, and the rate constants and $t1/2$ were calculated. One-way ANOVA was performed to compare the rate constants and $t1/2$ between strains, followed by the Dunnett test for pairwise comparison with MC4100 using GraphPad Prism version 10.0.2 software. $p < 0.05$ was considered statistically significant.

## Purified OM sample visualization via TEM

The purified OM samples (containing 10 μM BAM) were loaded onto 15 sec glow-discharged Cu Formvar/Carbon 300 mesh grids. After a 1 min incubation, the grids were stained with 1% uranyl acetate solution. The negative stained TEM images were collected using a FEI Tecnai T12 electron microscope.

## Analysis of in vivo OMP levels during late log phase

Single colonies of MC4100, NR698 (*lptD*4213), TN101(*mlaA*-), TN102 (*pldA*-) and TN103 (*mlaA*- *pldA*-) transformed with pJH114 were grown overnight in LB + MC/ampicillin at 37 °C at 250 rpm and used to inoculate fresh cultures at $OD_{600}$ = 0.05. The expression of BAM was induced at $OD_{600}$ ~ 0.8 by adding 0.4 mM IPTG for 45 min. Subsequently aliquots were removed from the cultures and centrifuged at 10,000 x *g* for 5 min at 4 °C. The cell pellets were resuspended in BugBuster® master mix (Millipore, catalog number 71456), incubated at room temperature for 15 min, and then mixed with 2x SDS-PAGE loading buffer. After samples were heated at 98 °C for 10 min, 10 µL of each sample (0.04 $OD_{600}$ equivalents) was subjected to SDS-PAGE. OMPs were detected by Western blotting and the OMP signal intensities were quantitated using ImageJ (version 2.14.0). The signal intensity of each OMP in MC4100 was set to 1.0 and used to normalize the signal intensities of the same OMP in the mutant strains. The averaged normalized intensities were plotted using GraphPad Prism (version 10.2.3).

## Characterization of strain NR698

The bile salt sensitivity of NR698[79] and other strains was assessed by the ability of the cells to grow on violet red bile glucose agar (VRBGA). *E. coli* strains transformed with pJH114 were streaked directly onto VRBGA plates from glycerol stocks and incubated overnight at 37 °C. In parallel, single colonies of MC4100 and NR698 transformed with pJH114 were grown overnight in LB + MC/ampicillin at 37 °C at 250 rpm and used to inoculate fresh cultures at $OD_{600}$ = 0.05. The cells were grown to $OD_{600}$ ~ 0.8 and centrifuged at 10,000 x *g*, 5 min at 4 °C. The cell pellets were resuspended in BugBuster® master mix (Millipore, catalog number 71456), incubated at room temperature for 15 min, and then mixed with 25 mM DTT and 2x SDS-PAGE loading buffer. After heating samples at 98 °C for 10 min, 5 µL of overnight culture samples (2.5 $OD_{600}$ equivalents) and 10 µL of stationary phase culture samples (0.8 $OD_{600}$ equivalents) were subjected to SDS-PAGE, and LptD and $LptD_{\Delta330-352}$ were detected by Western blotting.

## Generation of whole cell and purified OM samples for OMP, LPS and lipidomics analysis

Four independent cultures of each *E. coli* strain transformed with pJH114 were grown in 150 mL LB + MC and BAM expression was induced for 45 min as described above. At the end of the induction period, the $OD_{600}$ was recorded and two 5 mL aliquots of each culture were pelleted as whole cell (WC) samples and 100 mL of each culture was used to purify native OMs using the sarkosyl method described above. All of the purified OMs were divided into two equal aliquots, and the samples were flash frozen and stored at −80 °C. One set of aliquots from the WC and purified OM samples from each strain was used for lipidomics analysis while the other set of aliquots was used to measure LPS and OMP levels.

## Analysis of OMP and LPS levels in WC samples and purified native OMs used in lipidomics analysis

The cell pellets from the WC samples were resuspended in 20 mM Tris, pH 8.0 and aliquots of cell suspensions were normalized to 10 $OD_{600}$ equivalents/ml by mixing with 4 x Fluorescent compatible sample buffer (ThermoFisher, catalog number LC2570) while the purified OMs were diluted 100-fold by mixing with the same sample buffer. The samples were heated at 98 °C for 10 min and 10 µL from each WC sample (0.1 $OD_{600}$ equivalents) and OM (0.12-0.14 $OD_{600}$ equivalents from the original culture) were subjected to SDS-PAGE. OMPs and LPS were detected by Western blotting. The OMP signal intensities were quantitated using ImageJ and the mutant signal intensities were normalized to the MC4100 signal intensities as described above. The averaged normalized intensities were plotted using GraphPad Prism (version 10.2.3).

## Sample preparation for lipidomics analysis

For all liquid chromatography mass spectrometry (LCMS) methods LCMS-grade solvents were used. For lipidomics method development and analysis, ~5 × 10^9 bacteria or native OMs from ~5 × 10^10 of bacteria were immersed in 0.4 mL ice-cold methanol (Fisher Scientific, catalog number A456). Water (0.4 mL, Fisher Scientific, catalog number W6) and chloroform (0.4 mL, Fisher Scientific, catalog number C606SK4) were then added to each sample. Samples were shaken for 20 min under refrigeration and centrifuged at 16,000 x *g* for 20 min at 4 °C. A portion of the bottom (organic) phase (330 µL) was collected and dried down under vacuum. Samples were resuspended in an equivalent volume of 5 µg/mL butylated hydroxytoluene in 6:1 isopropanol: methanol. A composite quality control and method development sample was constructed by mixing a portion of the WC samples from all examined strains.

## PL profiling by Liquid Chromatography Tandem Mass Spectrometry (LC-MS/MS)

LCMS grade water (Fisher Scientific, catalog number W64), acetonitrile (Fisher Scientific, catalog number A9554), ammonium acetate (Fisher Scientific, catalog number A11450), and acetic acid (Fisher Scientific, catalog number A11350) were used for PL profiling by LC-MS/MS. All lipids were separated by headgroup class with a Waters XBridge Amide column (2.5 µm, 3 mm × 100 mm) on a LD40 X3 UHPLC (Shimadzu) using a 9 min binary gradient from 5% 12 mM ammonium acetate in water pH 7.5 in acetonitrile to 50% 12 mM ammonium acetate in water pH 7.5 in acetonitrile. In our method, all lipids were detected in negative polarity on a 7500 QTrap mass spectrometer (AB Sciex Pte. Ltd.) using a multiple reaction monitoring (MRM) strategy based on the hydride [M-1]- parent ion (or the dihydride $[M-2H]^{-2}$ for cardiolipin) and a daughter fatty acid ion for each lipid target. For diacyl PLs, the higher mass fatty acid ion was used when the fatty acids were not equivalent. For cardiolipin, lipid/fatty acid target pairs were calculated so that for each isobaric pool of cardiolipins with the same total number of acyl carbons and degrees of unsaturation, a signal was prepared for each potential fatty acid that could occupy a position in the molecule. Thus, the acyl content for each isobaric pool of cardiolipins could be measured but individual, isomerically pure cardiolipin species could not be quantified with confidence.

PL classes present in *E. coli* were determined based on a consensus of reports in the literature[82–84]. The classes that we targeted include PE, PG, CL, LPE, LPG, and free fatty acids (FA). Theoretical MRMs were first constructed for each lipid class for all possible combinations of fatty acids measured previously in *E. coli* via fatty acid methyl-ester (FAMES) analysis or suggested by previous untargeted lipidomic profiling[85–87]. Based on those previous reports our list included FA12:0, FA14:0, FA14:1, FA15:0, FA16:0, FA16:1, FA17:0, FAcy17:0, FA18:0, FA18:1, FA19:0, FAcy19:0 and FA20:0. It should be noted that a similar method was developed by Berezhonoy *et al.* (2022), but the details of the method were not made available and the CL series was not included[82]. The theoretical method, including all possible acyl chain combinations, was tested by injection of a method development mixed sample featuring portions of all strains examined here. MRM presence was initially assessed without scheduling to establish adequate retention time windows for each lipid class. Due to retention time coelution between PGs and CLs, certain candidate CL signals could not be resolved from potential isotope interference with PG signals. Likewise, all PG signals could not be fully isolated from CL in-source degradation. Mixed signals that could originate from either PG or CL were maintained in both the CL and PG series. If conclusions from method use relied on individual lipid signals and not class-wide behavior, it would have been essential to cross-check signals for potential lack of specificity. Cardiolipin distribution was compared to results described in previous reports to ensure consistency with the expected hierarchy in *E. coli*[107]. Following test injections, the theoretical method was refined to

include only targets with a signal-to-noise ratio greater than 3, and retention time scheduling was implemented to maximize sensitivity and datapoints per peak. Method details, including instrument parameters and specific MRMs, are included in the Source Data file.

### Lipidomics LC-MS/MS data processing

All signals were integrated using SciexOS 3.1 (AB Sciex Pte. Ltd.). Signals with >50% missing values were discarded, and remaining missing values were replaced with the lowest registered signal value. Signals with an average intensity, after total sum normalization, across the dataset below 30,000 counts were excluded. Signals with a Quality Control (QC) coefficient of variance >30% were discarded. Filtered raw data are included in Supplementary Data 2.

### Bioinformatics and calculations

All statistical analyses were performed using MarkerView (AB Sciex) or in-house developed R code (version 4.4.1; 2024-06-14). Data visualization was performed with the ggplot2 package in R or GraphPad Prism 10. The matrix barplots presented in this paper were generated using the publicly available mbplot R package which was developed in-house[108]. Principal Component Analysis (PCA) was performed on Pareto-scaled data in MarkerView. The $t$-test function from the rstatix R package was used to measure mean lipid level differences between OM and WC fractions. The rstatix ANOVA function was used for one-way comparison of lipid level differences between mutant strains. To reduce Type I error (false positives) from multiple comparisons, resulting $p$-values were adjusted, where indicated, using the rstatix adjust_$p$-value function with the Benjamini-Hochberg method. Average fold changes for lipid classes were used to estimate an expected OM PE to PG + CL ratio for comparison to published results. Based on a previous report, an IM molar ratio of 3.5 was assumed for PE/(PG + CL)[53]. Furthermore, we assumed a distribution of 33% of total PLs localized to the OM, due to the expected LPS occupancy of the outer leaflet of the OM. The average fold changes by lipid class for lipids that significantly varied between the OM and WC in our data was 1.75 for PE and 0.41 across PG and CL. These average fold changes by lipid class were combined to yield an estimated fold change in the OM PE/(PG + CL) ratio of 3.72 compared to the WC lipidome. Using the assumptions described above, the PE and PG + CL fractions in the OM were calculated, yielding an expected OM PE/(PG + CL) of 18.88, which is within the range of values observed in Lugtenberg & Peters[53].

### Generated images

Appropriate crystal structures were obtained from the Protein Data Bank, www.rcsb.org[109] and the protein structure images were created using PyMol Molecular Graphics System version 2.1[110]. The lipid bilayers in Fig. 1 were generated using BioRender.

### Reporting summary

Further information on research design is available in the Nature Portfolio Reporting Summary linked to this article.

## Data availability

All of the lipidomics data are provided in Supplementary Data 2. The lipidomics data are also available at figshare (https://doi.org/10.6084/m9.figshare.30780602.v1). The accession codes used in this study are all from the Protein Database (PDB): 6R7L, 1M5Y, 5D0O, 4Q35, 1QD6. 5NUP, 5UWA, 6ZY3, and 3SLJ. Source data are provided with this paper. MS strategies were targeted and signals were analyzed based on known lipid fragmentation conventions. Consequently spectra were not acquired for individual lipids. Method details including instrument parameters and specific MRM's are included in the source data file. Source data are provided with this paper.

## Code availability

The code used in this study is available at Zenodo (https://doi.org/10.5281/zenodo.14755324).

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

## Acknowledgements

We thank Natividad Ruiz (Ohio State University) for providing strain NR698 as well as her unpublished observations, Tom Silhavy (Princeton University) for providing anti-LptD, and Nidhi Kundu (Laboratory of Molecular Biology, NIDDK) for providing technical support and assistance with the TEM imaging. The TEM images were collected at the NIDDK Cryo-EM core facility. We would also like to thank Russell Bishop (University of Toronto) and Zhixin Lyu (Genetics and Biochemistry Branch, NIDDK) for providing insightful comments on the manuscript. This work was supported by the Intramural Research Programs of the National Institute of Diabetes and Digestive and Kidney Diseases and the National Institute of Allergy and Infectious Diseases within the National Institutes of Health (NIH). The contributions of the authors were made as part of their official duties as NIH federal employees, are in compliance with agency policy requirements, and are considered Works of the United States Government. However, the findings and conclusions presented in this paper are those of the authors and do not necessarily reflect the views of the NIH or the U.S. Department of Health and Human Services.

## Author contributions

This study was designed and analyzed by T.D.N. and H.D.B. All of the experimental work, except the lipidomics, was performed by T.D.N. The lipidomics method was developed and lipidomics data was acquired by N.T.B. and B.S. The lipidomic data were analyzed, and corresponding figures were generated by I.S.L., N.T.B., and B.S. The manuscript was written by T.D.N., B.S., and H.D.B.

## Funding

## Competing interests

The authors declare no competing interests.
