## [Transparent Peer Review file · Nature Communications]

Phospholipid composition strongly affects the assembly of β barrel proteins into purified bacterial outer membranes

Corresponding Author: Dr Harris Bernstein

Version 0:

Reviewer comments:

Reviewer #1

(Remarks to the Author)

This study investigates the influence of outer membrane (OM) lipid composition on the assembly of β -barrel outer membrane proteins (OMPs) into purified bacterial OMs. Although the bacterial OM is a complex structure, prior in vitro studies have demonstrated OMP assembly using isolated components, such as naturally shed outer membrane vesicles (OMVs) from Gram-negative bacteria. Additionally, de novo synthesized OMPs have been shown to incorporate, albeit slowly, into crude *E. coli* microsomal membranes (EMMs) purified in the presence of EDTA, which removes lipopolysaccharides (LPS) from the OM. However, a significant limitation of these earlier approaches is that the membrane vesicles used do not faithfully recapitulate the asymmetric lipid composition or the native OMPs found in authentic OMs.

To address these limitations, the authors have developed an improved in vitro assay using purified native OM fractions to examine the activity of the β -barrel assembly machinery (BAM) in a more physiologically relevant context. They demonstrate that BAM present in these purified membranes can assemble OMPs, and that this assembly is influenced by specific features of OM lipid composition. Through phospholipidomic analysis of the *E. coli* OM, they further show that mutations disrupting OM lipid homeostasis alter the ratio of OM to whole-cell phospholipids and lead to global shifts in phospholipid acyl chain length.

The strength of this work lies in the development of a system that uses the native *E. coli* OM and can serve as a robust platform for studying OMP biogenesis in vitro. This method has potential applications in identifying in vivo inhibitors of BAM activity and in gaining deeper mechanistic insights into its function. Furthermore, the authors' comprehensive lipidomics approach, resolving individual lipid species, provides valuable information on how lipid composition governs OM assembly. The experimental design is sound with some exceptions and the study addresses a complex biological process.

However, a major concern is the potential disparity between the in vitro results and the physiological conditions in vivo, which raises questions about the assay's ability to distinguish direct from indirect effects on BAM function. Additionally, the study focuses only on a limited set of mutants (*mIaA*, *pIaA*, and *lptD4213*) and does not consider other systems involved in OM lipid homeostasis. Consequently, the conclusions drawn are limited to the mutants analyzed. Moreover, some key experimental controls are lacking, and additional experiments are needed to substantiate the main claims (see comments below). These issues somewhat constrain the broader impact and generalizability of the study.

Major Comments:

1. The study does not consider the inclusion of additional lipid transport systems mutants such as Tol-Pal complex, which is implicated in the retrograde transport of bulk phospholipids (PLs) from the OM to the inner membrane (IM). Tol-Pal mutations lead to PL accumulation in the OM and slower retrograde transport. Similarly, no analysis was performed on AsmA-like proteins (e.g., TamB, YhdP, YdbH), which are essential collectively but could still be evaluated using depletion strains. These strains exhibit elevated PL levels at the IM and could provide additional insight into lipid-driven effects on BAM activity.
2. The authors analyzed mutants in *LptD* (an OMP), *MlaA* (an OM lipoprotein), and *PldA* (an OM phospholipase), and observed varying effects on β -barrel protein assembly. However, it remains unclear whether the observed effects on BAM-mediated OMP assembly result directly from lipid compositional changes or are secondary to broader OM perturbations. Notably, TEM images (Figure S9) reveal heterogeneity in vesicle size and ruptured fragments, which could indirectly impact BAM function. Including a negative control such as an OMP mutant unrelated to OM lipid homeostasis would help determine whether the observed effects on BAM activity are lipid-specific or due to general OM perturbations. This is important to rule out any indirect effects of OM mutations.
3. The authors only examine the *mIaA* mutant, despite the *Mla* system comprising multiple components, including the

periplasmic protein MlaC and the inner membrane complex MlaDEFB. Since all components contribute to retrograde PL transport, it is important to test whether mutations in other parts of the Mla pathway also impact BAM activity. Including additional mla mutants would strengthen conclusions about the relationship between Mla function and BAM activity. Overall, analysis with respect to the Mla system is incomplete.

4. The manuscript does not sufficiently establish the specificity of the effects observed in mlaA, pldA, and lptD mutants on BAM function. Complementation assays by reintroducing wild-type copies of these genes into the respective mutants should be performed to confirm whether BAM activity in assembling EspPΔ5' is restored. Additionally, TEM analysis of the complemented strains would help verify if vesicle morphology returns to a more uniform, wild-type-like state.

5. The resolution and labeling of Figures 5a–c is suboptimal and hinder interpretation. Consider presenting a smaller, clearer subset of data in the main text and moving the remainder to supplementary figures. Improving figure quality is essential for clear data communication.

6. Figure S8 lacks quantitative analysis. Qualitatively, BamA levels appear higher in NR698 and lower in TN103. FepA levels seem lower in TN103; LamB levels are reduced in TN102 and TN103, while FadL is elevated in NR698 and TN101. OmpT appears in increased TN103. These observations require densitometric quantification to support conclusions about protein expression patterns.

Minor Comments:

1. Line 74: Update the description of BamD to reflect that its essentiality is conditional and varies with substrate.

2. Line 103: Include YdbH alongside TamB and YhdP for completeness when referencing AsmA-like anterograde lipid transporters.

3. Figures 2d and S3 should display data as "% fold change" for clarity.

4. Clarify whether lipid content was normalized between induced and non-induced samples or if only BAM concentrations were equalized. Unequal lipid amounts could account for differences in observed assembly efficiency.

5. Figure 4c: Improve image quality and clarify the numeric labels, which are currently difficult to read.

6. Including a TLC image of PE, PG, and CL across mutant strains would provide an intuitive visual validation of the mass spectrometry data and strengthen the lipidomic conclusions.

Reviewer #2

(Remarks to the Author)

Folding and inserting β -barrel membrane proteins is a fundamental biological process. In the Gram-negative bacterial outer membrane, the β -barrel assembly machine (BAM) catalyzes this process for outer membrane proteins (OMPs), however the precise mechanistic details of OMP folding by BAM have yet to be resolved. In their manuscript "Phospholipid composition strongly affects the assembly of β -barrel proteins into purified bacterial outer membranes", Dr. Harris Bernstein and colleagues describe a novel approach for studying OMP folding in vitro and systematically analyze the phospholipid composition of the outer membrane. Their observations lead to novel insight about the impact of differing outer membrane phospholipid compositions on BAM activity.

This is an extremely well-written paper, the experiments (including critical controls) are elegantly designed and described, and the conclusions are consistent with the reported results. Both the techniques and the findings of this work will be of high interest to the readership of Nature Communications, especially those interested in microbiology, membrane biology, protein folding, and antibiotic discovery, and have the potential to be generally impactful across multiple avenues of research. I have no major issues that would preclude publication of this work. Below I have provided some specific questions and comments that the authors can consider.

- Line 45 – The authors highlight *Acinetobacter baumannii* and *Pseudomonas aeruginosa* as critical Gram-negative pathogens (which they are), but members of the Enterobacteriaceae, including *E. coli* and *K. pneumoniae*, are just as critical in terms of the number of patients affected and antibiotic resistance. It is a good opportunity to highlight these for the broad audience of Nature Communications.

- Line 168 and Figure S1 and S5 – The reported data clearly show equivalent levels of BamA, BamB, and BamD, but leave open the possibility that BamC and BamE levels could differ. Is there any possibility that different levels of BamC and/or BamE account for any subsequent difference in OMP folding activity?

- Line 210 – Is there an obvious reason why less SurA is required compared to previously reported in vitro folding assays? Could this be explained by access (e.g., the OMV-based assay necessitated getting SurA inside the vesicles)?

- Line 217 and Figure 2c – The discussion notes that how the substrate is produced can explain the inability of Skp and/or DegP to facilitate folding of EspPΔ5'. Could the role of these chaperones also be substrate dependent (e.g., could they play a role with different OMPs)? More generally, have you measured the folding of trimeric OMPs with your in vitro assay?

- Line 237 – The observation that BAM density can dictate activity is intriguing. Did you test whether this deficit in BAM activity in the uninduced sample be overcome by increasing the amount of substrate and/or SurA?

- Figure 2c – It looks like there is less overall EspPΔ5' in some lanes (e.g., DegP, SurA+Skp, SurA+DegP). Is there an explanation for this (e.g., degradation)?

- Line 273 – More of an observation than a question, but you point out that the t1/2 for assembly are slower for TN102, but is

there any reason to think that the ~3.5-minute t1/2 versus the ~2.5-minute t1/2 for the other three strains is biologically meaningful? Perhaps not given that Figure S8 indicates that the OMP levels (except for, perhaps, FepA) are more or less the same (at least under the tested growth conditions).

- Line 399 and Figure 6 – You note that the effects of lacking MlaA and PldA are synergistic. Is this the right description? TN103, I think, looks like a distinct pattern from each of the individual mutants, but perhaps I am misinterpreting this?

- A more general question: did you run lipidomic analysis of outer membranes isolated using the sucrose gradient approach? The reported comparisons with the whole cell preparations are extremely helpful, but one point that remains is whether the sarkosyl detergent has any added effect on the isolated lipid composition. Assessing the phospholipid composition in a detergent-free preparation could be informative for this question.

- A minor suggestion to help with readability throughout - The authors could consider adding the relevant genotypes whenever the strains are noted in the text (for example: NR698 lptD4213, TN101 Δ m1aA, TN102 Δ pldA, TN103 Δ m1aA Δ pldA) so the reader does not have to continually refer back to the original strain descriptions.

- Again, more of a comment than a question, but I am very interested to see future applications of this OMP folding assay and lipidomic measurements performed under more perturbative conditions that might reveal additional effects of PldA and MlaA and insight into the mechanism of BAM.

Reviewer #3

(Remarks to the Author)

This manuscript by Nilaweera et al present data that raises some interesting questions about the role of phospholipids in the outer membrane (OM). Following optimisation of conditions for Bam expression in OM fractions they analyse activity as judged by OMP folding. This contrasts previously used methods of proteoliposomes, providing an alternative method for functional analysis of Bam. Similar to previously reported data results showed SurA is essential for folding of EspP (primary read out of activity in manuscript). Nilaweera and team observe that induction of Bam during production of OM fractions is more active than the same concentration of Bam from a uninduced sample. This aligns with other data in the literature that shows OMP clusters to be important for membrane stability and organisation. I found the latter sections to become more noteworthy as Bam activity was compared between mutant strains with impaired lipid transport. PldA deletions had the most pronounced effect on Bam activity reducing folding from 37% in WT to 7%. MlaA and LptD deletions had a weaker effect reducing the rate of activity without impacting amount of protein folded as significantly. Elegant LC-MS/MS experiments were developed to study the proteolipidomes of the mutants. This showed that PldA mutants had an OM that resembled the IM PL profile, whereas Mla was similar to WT cells. This supports the previous idea that each system has distinct roles in the mechanism for removal of PL from the OM.

I found the experimental procedures to be largely robust, and the material, particularly the LC-MS/MS, was both novel and interesting. The data clearly demonstrate that phospholipids influence Bam activity within this in vitro system. The authors appropriately acknowledge that further investigation is needed to elucidate the in vivo relevance of these findings. It remains to be determined whether the phospholipids themselves are directly involved in protein interactions, or if the observed effects are due to lipid-induced changes in membrane fluidity and disruption of outer membrane organisation and stability. I have outlined several points below where further clarification would enhance the manuscript.

Further discussion and clarification required:

- Line 238-239 and 435-442: Native levels of Bam provide sufficient activity to fold OMPs in vivo. Could the phenomenon that is observed here be related to observations in this paper: <https://www.pnas.org/doi/10.1073/pnas.2414725122> That mentioned OMPs are largely mobile so to create the clusters Bam has to be concentrated in these precincts to maintain OM organisation and lipids can insert anywhere.
- Line 464-466: could the authors comment on inclusion or exclusion of the possibility that the specific PL enrichment causes a more fluid membrane where the organisation of the proteins is not maintained. Is it this organisation that impacts activity rather than the PL themselves. Neighbouring OMPs creates a more rigid membrane that aid folding rather than lipids, particularly phospholipids that will create a very fluid environment, like the IM, and lower insertion rates due to a lack of supporting structure around Bam.
- Line 415-416: Could the authors clarify why membrane crosstalk is concluded as the primary explanation, rather than the alternative that Mla mutations have a weaker effect because phospholipids from the inner membrane are efficiently removed by the Pld? When the Pld is absent, Mla that might remove phospholipids with specific acyl chain lengths cannot fully compensate, leading the OM to adopt a phospholipid profile resembling that of the IM. Rather than crosstalk, this may reflect distinct, complementary roles of the two phospholipid removal systems, each influencing the lipid profile in different ways.

Minor points:

- A general point: the strain names are hard to remember what the mutation is, using a short hand such as Δ MlaA would make it easier for the reader.
- Line 37: It is well established that MlaA and PldA have distinct roles in phospholipid removal from the OM. It may be helpful to revise the wording to reflect their non-redundant functions more clearly.
- Line 54-56: Whilst OMPs are not conventional alpha helical membrane proteins, their conservation and high copy numbers in these bacteria would suggest that they are perhaps not that "unusual". Revised wording could be beneficial.
- Line 89-90: Please check the wording here as it is unclear. Suggest removing the "however".

- Line 92: While I appreciate the authors' emphasis on the importance of membrane asymmetry and acknowledge that certain OMPs can be deleted without completely inhibiting growth, there are important caveats. For instance, components such as Bam and Lpt are essential, and cells lacking core OMPs exhibit significant stress or impaired viability. I suggest rephrasing the sentence to reflect these nuances.
- Line 121-122: OMVs should have a fully intact OM including asymmetry and OMP profile representative of the region of the cell from which it budded off.
- Line 130: The authors state that outer membrane (OM) fractions provide a more representative view of the OM landscape than outer membrane vesicles (OMVs). However, the images of OM fractions prepared using Sarkosyl show the formation of small, vesicle-like structures. Given that OMVs retain intact membranes, one could argue that they share similar structural features. The challenge with vesicles lies in the uncertainty regarding their origin, are they derived from specific regions of the cell envelope and do they contain outer and/or inner membrane components. A potentially stronger and more defensible point would be that the OM fraction may offer broader sampling of the OM across the entire cell surface, thereby providing a more comprehensive overview.
- Line 131 vs 135: OM fractions referred to native OMs got confusing on the way through. Further clarification, native OMs is referring to the intact nature through use of Sarkosyl not native meaning wild-type vs mutant.
- Line 174: Will the observed effect be due to overall OM functionality, or specifically a result of Bam complex activity? Please rephrase to clarify this distinction.
- Line 184/601-602: Much of the analysis and comparison presented relies on accurately determining the Bam concentration within the samples. While western blotting is one of the few feasible methods for estimating protein levels in a complex mixture like the OM fraction, it is often limited by variability and the quality of the antibodies used. It would be helpful if the authors could include a control blot in the supplementary material, showing OM fractions with Bam at a few specified concentrations across samples (e.g., induced vs. non-induced conditions). This would help demonstrate the reliability of the measurements since 1–2 μM Bam is relatively low, and showing this would support the observed differences in downstream folding assays.
- Line 186: How many lysine residues are modified per EspP molecule, and is the extent of modification consistent across all molecules? I ask this to assess whether the modification might influence folding kinetics. If the degree of BODIPY labeling on lysines is uniform across samples, that would help support the validity of the folding comparisons. Was this assessed or controlled for?
- Figure 2: I generally really like the figures in this manuscript. I just wonder if 2a and 2b might be better placed in the supplementary material?
- Figure S3: Could the authors add to the figure caption to define what the control sample is?
- Line 210: Looking at the figure it could be argued that anywhere from 4-10 μM is optimal, could the authors comment on how significant is the difference is between these points to clarify why 8 μM was defined as optimal
- Line 258: How much peptidoglycan would be expected to remain in the preparation? If present, it's worth noting that mature peptidoglycan has been shown not to significantly affect activity. Therefore, the observation that lysozyme treatment has little impact may not be unexpected. Clarifying this point could help contextualise the result.
- Line 265: NR698 "appears" to fold more EspP than TN101 but less than wild-type in figure 4. To me this suggests that the MlaA mutation has a more pronounced effect than the LptD mutation. Combined with the time course in figure S7 NR698 looks more comparable to WT whilst MlaA lags until time point 30min. Similarly in Fig 4c TN101 looks to lag more after that 50% maximum assembly point. Are the approximate time differences and % of EspP folded significant and if so could the authors clarify on how this was calculated/concluded. It will help to recapitulate what is observed at first glance of the figures with the text.
- Line 275-276: TN103 appears to show impaired Bam function similar to TN102, could the authors clarify this within the text here?
- Fig S9/Line 290: Could the authors speculate why LPS-deficient strain NR698 produce smaller vesicles compared to the controls? Understanding the underlying cause would provide valuable insight into membrane dynamics in these mutants.
- Fig S10 and Fig S8. There appears to be some variability in the levels of outer membrane proteins (OMPs) such as BtuB, FepA, LamB, FadL, and OmpT in the purified OM samples relative to whole cell. Could the authors comment on the repeated data in WC samples and why S8 was not done with OM fractions that were analysed throughout? Specifically, Line 284–286 states that OMP levels are consistent, which seems to hold true in WC samples, but may not fully apply to purified OM fractions. Given that mutations leading to increased phospholipid content likely impair the OM and promote OMP shedding during detergent treatment, it is surprising that TN102 with pronounced effects on Bam function appears to have higher OMP levels in OM samples than TN101.
- Line 294: It is not convincing at the resolution shown in images that there is a "double-layer" to vesicles. Higher magnification and resolution would be required to make this convincing.
- Line 348-349: can the authors comment on why LPS is consistent across all strains even in the LptD insertion machinery mutant?
- Line 447: could authors expand on how number of active BAM complexes is discerned from consistent number of Bam complexes with reduced rate?

REPLY TO REVIEWERS' COMMENTS (all changes in the manuscript are highlighted)

Reviewer #1 (Remarks to the Author):

This study investigates the influence of outer membrane (OM) lipid composition on the assembly of β -barrel outer membrane proteins (OMPs) into purified bacterial OMs. Although the bacterial OM is a complex structure, prior in vitro studies have demonstrated OMP assembly using isolated components, such as naturally shed outer membrane vesicles (OMVs) from Gram-negative bacteria. Additionally, de novo synthesized OMPs have been shown to incorporate, albeit slowly, into crude E. coli microsomal membranes (EMMs) purified in the presence of EDTA, which removes lipopolysaccharides (LPS) from the OM. However, a significant limitation of these earlier approaches is that the membrane vesicles used do not faithfully recapitulate the asymmetric lipid composition or the native OMPs found in authentic OMs. To address these limitations, the authors have developed an improved in vitro assay using purified native OM fractions to examine the activity of the β -barrel assembly machinery (BAM) in a more physiologically relevant context. They demonstrate that BAM present in these purified membranes can assemble OMPs, and that this assembly is influenced by specific features of OM lipid composition. Through phospholipidomic analysis of the E. coli OM, they further show that mutations disrupting OM lipid homeostasis alter the ratio of OM to whole-cell phospholipids and lead to global shifts in phospholipid acyl chain length.

The strength of this work lies in the development of a system that uses the native E. coli OM and can serve as a robust platform for studying OMP biogenesis in vitro. This method has potential applications in identifying in vivo inhibitors of BAM activity and in gaining deeper mechanistic insights into its function. Furthermore, the authors' comprehensive lipidomics approach, resolving individual lipid species, provides valuable information on how lipid composition governs OM assembly. The experimental design is sound with some exceptions and the study addresses a complex biological process.

However, a major concern is the potential disparity between the in vitro results and the physiological conditions in vivo, which raises questions about the assay's ability to distinguish direct from indirect effects on BAM function. Additionally, the study focuses only on a limited set of mutants (mIaA, pIdA, and lptD4213) and does not consider other systems involved in OM lipid homeostasis. Consequently, the conclusions drawn are limited to the mutants analyzed. Moreover, some key experimental controls are lacking, and additional experiments are needed to substantiate the main claims (see comments below). These issues somewhat constrain the broader impact and generalizability of the study.

We thank the reviewer for his/her positive remarks. Our response to the reviewer's concerns appears below.

Major Comments:

1. The study does not consider the inclusion of additional lipid transport systems mutants such as Tol-Pal complex, which is implicated in the retrograde transport of bulk phospholipids (PLs) from the OM to the inner membrane (IM). Tol-Pal mutations lead to PL accumulation in the OM and slower retrograde transport. Similarly, no analysis was performed on AsmA-like proteins (e.g., TamB, YhdP, YdbH), which are essential collectively but could still be evaluated using depletion strains. These strains exhibit elevated PL levels at the IM and could provide additional insight into lipid-driven effects on BAM activity.

We completely agree with the reviewer. We would like to note, however, that although we are beginning to conduct the experiments that he/she recommends, we believe that they are beyond the scope of the present study and we plan to include the results in a separate manuscript. Our goal was

simply to show that mutations that affect the PL profile of *E. coli* affect the activity of BAM under our experimental conditions. A great deal of work was required to characterize and optimize the novel OMP assembly assay that we describe, to develop the method to perform the lipidomics analysis, and to analyze the lipidomics data. It would take at least many months (if not more than a year) to complete an analysis of the effects of mutations in the Tol-Pal complex (which have been reported to have pleiotropic effects on cell physiology) and multiple genes that encode AsmA-like proteins. The manuscript already contains a relatively long Discussion section and 14 supplementary Figures. It is likely that many additional issues would arise from an analysis of additional mutant strains, our manuscript would grow in length and complexity, and it would overwhelm readers. Our basic message would likely also get lost in the shuffle.

2. The authors analyzed mutants in LptD (an OMP), MlaA (an OM lipoprotein), and PldA (an OM phospholipase), and observed varying effects on β -barrel protein assembly. However, it remains unclear whether the observed effects on BAM-mediated OMP assembly result directly from lipid compositional changes or are secondary to broader OM perturbations. Notably, TEM images (Figure S9) reveal heterogeneity in vesicle size and ruptured fragments, which could indirectly impact BAM function. Including a negative control such as an OMP mutant unrelated to OM lipid homeostasis would help determine whether the observed effects on BAM activity are lipid-specific or due to general OM perturbations. This is important to rule out any indirect effects of OM mutations.

As we show in our manuscript, the mutations did not significantly affect the levels of eight model OMPs *in vivo* at different growth phases (Fig. S9) or the level of the same OMPs or LPS in purified native OMs (Fig. S11). Although the TEM analysis showed that the native OMs isolated from the wild-type and mutant strains differed in morphology, we did not see a correlation between the degree to which the morphology of the mutant vesicles differed from the wild-type vesicles and the effect of the mutations on OMP assembly. Taken together, these results strongly suggest that the mutations do not broadly perturb the OM. For that reason the changes in the PL composition—which do correlate with the effect of the mutations on OMP assembly—are the most likely factors that determine the efficiency of OMP assembly. It is impossible to know, however, if the effect of the changes in PL composition is “direct” or “indirect”, and we never claim that the effect is “direct”. One major problem is that there is no simple definition of “direct”. Would a change in the level of a lipid that binds to BamA be a “direct” effect, or would this term also apply to changes in the local lipid environment?

In principle, we agree that it would be worthwhile to include a negative control such as native OMs derived from a strain that lacks an OMP that does not play a role in maintaining OM lipid homeostasis. It is unclear, however, which OMP we should mutate, because we do not know if the absence of a generic OMP might in itself alter the PL composition of the OM or create a broader OM perturbation that would complicate the interpretation of the data. In this regard we should mention that both PldA and MlaA are produced at relatively low levels. In fact, we cannot detect PldA on a Western blot unless we clone the *pldA* gene into a plasmid and overexpress it (see PMID 28941249). For that reason it seems unlikely that simply removing PldA and/or MlaA would perturb the OM to the same degree as removing an abundant OMP like OmpC.

3. The authors only examine the *mlaA* mutant, despite the Mla system comprising multiple components, including the periplasmic protein MlaC and the inner membrane complex MlaDEFB. Since all components contribute to retrograde PL transport, it is important to test whether mutations in other parts of the Mla pathway also impact BAM activity. Including additional *mla* mutants would strengthen

conclusions about the relationship between Mla function and BAM activity. Overall, analysis with respect to the Mla system is incomplete.

Our main goal was to determine if changes in the PL profile of the OM that occur as a result of disrupting *mlaA* might affect BAM activity. That is to say, our goal was to focus on PL profiles rather than the Mla pathway per se. We agree that it would be reassuring if we found that the effects of disrupting *mlaA* and *mlaB* (or *C*, *D*, *E* or *F*) were similar, but what if the effects are different because cells respond differently to the loss of each component? We believe that the results that we have already obtained are sufficient to show a correlation between BAM activity and the PL composition of the OM, and that further explorations of the Mla pathway might lead to a sidetrack.

4. The manuscript does not sufficiently establish the specificity of the effects observed in *mlaA*, *pIdA*, and *IptD* mutants on BAM function. Complementation assays by reintroducing wild-type copies of these genes into the respective mutants should be performed to confirm whether BAM activity in assembling EspPΔ5' is restored. Additionally, TEM analysis of the complemented strains would help verify if vesicle morphology returns to a more uniform, wild-type-like state.

In principle this is an excellent idea, and we agree that complementation experiments are very useful in some experimental contexts. Most notably, complementation experiments can be used to show that a mutation does not affect the expression of downstream genes. We have found, however, that *mla::kan* does not significantly affect the level of FadL, which is encoded by a closely linked gene, and *pIdA::kan* is nowhere near a gene that encodes a relevant cell envelope or lipid biosynthetic protein. Furthermore, in our experimental system complementation experiments would be trickier to interpret than they might seem. The main problem is controlling the level of expression; because the level of MlaA and PIdA would presumably not match the level of the proteins in wild-type cells, the PL profiles would likely also differ from those of wild-type cells. It is also unclear how the overexpression of *mlaA* relative to other genes that encode components of the *mla* pathway would affect activity. Although complementation might partially restore or even enhance BAM function, significant differences in activity would be difficult to interpret.

5. The resolution and labeling of Figures 5a–c is suboptimal and hinder interpretation. Consider presenting a smaller, clearer subset of data in the main text and moving the remainder to supplementary figures. Improving figure quality is essential for clear data communication.

To address this concern, we have now increased the size of the data points and all of the labels (both the lipid and axis labels) in Fig. 5a and Fig. 5c to improve the visibility of the results. The data points and axis labels in Fig. 5b are already large and easy to see. We believe that the results shown in Fig. 5 are central to our study and should not be moved to the Supplement.

6. Figure S8 lacks quantitative analysis. Qualitatively, BamA levels appear higher in NR698 and lower in TN103. FepA levels seem lower in TN103; LamB levels are reduced in TN102 and TN103, while FadL is elevated in NR698 and TN101. OmpT appears in increased TN103. These observations require densitometric quantification to support conclusions about protein expression patterns.

To address the reviewer's concern we have performed three replicates of the experiment shown in Fig. S8 (now Fig. S9) and performed a quantitative analysis. Our quantitative analysis shows that there is a small amount of variation in the OMP levels that likely results from experimental error. The average levels of LamB and FadL in TN102 and TN103 are quite similar to the average levels in the wild-type

strain (MC4100). There was more variation in OmpT than some of the other OMPs, but the average level of the protein in the three replicates turned out to be slightly lower in TN103 than MC4100. As we note, we were unable to quantitate the levels of FepA because the signals were too low. Nevertheless, given that the average level of all of the other OMPs in all of the mutant strains was within ~20% of their average level in MC4100, it would be fair to say that the protein levels were very similar in all of the strains. Furthermore, given that the small differences in the OMP levels do not correlate with the phenotypes that we observed in our cell free assay, it would also be fair to conclude that the phenotypes are almost certainly not due to differences in the OMP composition of the native OMs that we purified.

We should also note that we removed the part of Fig. S8 in which we showed OMP levels in cells in stationary phase because after three replicates we found that there was significant fluctuation in the data that is difficult to interpret. Perhaps more importantly, the levels of OMPs in stationary phase is really not relevant to our study.

Minor Comments:

1. Line 74: Update the description of BamD to reflect that its essentiality is conditional and varies with substrate.

We have made the recommended change (see line 75).

2. Line 103: Include YdbH alongside TamB and YhdP for completeness when referencing AsmA-like anterograde lipid transporters.

We have made the recommended change (see line 105).

3. Figures 2d and S3 should display data as “% fold change” for clarity.

We believe that the reviewer is referring to Figs. 2E and S4 (now S5). For clarity, we have changed the label to “% folded” and we now describe how this value was calculated in the Methods section (see lines 641-642). We have also made the same change in Figs. S1D, S7 and S8.

4. Clarify whether lipid content was normalized between induced and non-induced samples or if only BAM concentrations were equalized. Unequal lipid amounts could account for differences in observed assembly efficiency.

We only normalized the concentration of BAM. Indeed it would be difficult to normalize both BAM and lipid content because one can normalize only one variable at a time. Given that we observed a higher level of BAM activity when the average concentration of BAM per vesicle was higher, it is difficult to imagine how unequal lipid amounts would account for the difference in activity. One can imagine that BAM might crowd out lipids that are required for activity, in which case we would have obtained the opposite result, but it is unclear how an excess of lipid would be deleterious. We have never obtained evidence that the protein to lipid ratio in proteoliposomes that contain purified BAM, which is inherently variable, significantly affects activity in our *in vitro* assays. Nevertheless, the reviewer raises a good point: when we overexpressed BAM, the protein and/or lipid composition of the OM might have changed. We tried to figure out how we might change the sentence on lines 244-246 to address this issue, but we concluded that to avoid confusing readers we should leave the sentence as is. The sentence is already appropriately vague in that we state that the results “suggest” that the density of

BAM affects activity and indicate that the “composition of native OMs” (which refers to a complex mixture of proteins and lipids) influences BAM activity.

5. Figure 4c: Improve image quality and clarify the numeric labels, which are currently difficult to read.

To address this concern we have rearranged the panels, relabeled the graph, and increased the size of the text in the box in part C.

6. Including a TLC image of PE, PG, and CL across mutant strains would provide an intuitive visual validation of the mass spectrometry data and strengthen the lipidomic conclusions.

We believe that the volcano plots, PC analyses, and heatmaps in Figs. 5 and 6 and Figs. S13 and S14 provide a clear picture of the MS data and provide much greater detail than TLC. As we note in the text (lines 343-345), our results are consistent with the results of TLC experiments conducted 50 years ago, and we believe that our conclusions would be only very modestly strengthened by replicating both the results and the methods of the earlier studies.

Reviewer #2 (Remarks to the Author):

Folding and inserting β -barrel membrane proteins is a fundamental biological process. In the Gram-negative bacterial outer membrane, the β -barrel assembly machine (BAM) catalyzes this process for outer membrane proteins (OMPs), however the precise mechanistic details of OMP folding by BAM have yet to be resolved. In their manuscript “Phospholipid composition strongly affects the assembly of β -barrel proteins into purified bacterial outer membranes”, Dr. Harris Bernstein and colleagues describe a novel approach for studying OMP folding in vitro and systematically analyze the phospholipid composition of the outer membrane. Their observations lead to novel insight about the impact of differing outer membrane phospholipid compositions on BAM activity.

This is an extremely well-written paper, the experiments (including critical controls) are elegantly designed and described, and the conclusions are consistent with the reported results. Both the techniques and the findings of this work will be of high interest to the readership of Nature Communications, especially those interested in microbiology, membrane biology, protein folding, and antibiotic discovery, and have the potential to be generally impactful across multiple avenues of research. I have no major issues that would preclude publication of this work. Below I have provided some specific questions and comments that the authors can consider.

We thank the reviewer for his/her positive and insightful comments.

- Line 45 – The authors highlight *Acinetobacter baumannii* and *Pseudomonas aeruginosa* as critical Gram-negative pathogens (which they are), but members of the Enterobacteriaceae, including *E. coli* and *K. pneumoniae*, are just as critical in terms of the number of patients affected and antibiotic resistance. It is a good opportunity to highlight these for the broad audience of Nature Communications.

We have made the suggested change.

- Line 168 and Figure S1 and S5 – The reported data clearly show equivalent levels of BamA, BamB, and

BamD, but leave open the possibility that BamC and BamE levels could differ. Is there any possibility that different levels of BamC and/or BamE account for any subsequent difference in OMP folding activity?

Most of our experiments (including those shown in Figs. S1 and S5 [now S6]) were performed under conditions in which all of the genes that encode BAM subunits were cloned into a plasmid to elevate their expression. We found previously that the BAM subunits are all produced at the same level under these conditions, and there is no obvious reason to suspect that BamC and or BamE would be selectively degraded. Even if they were, available evidence suggests that neither protein plays a significant role in OMP assembly, so it seems unlikely that a reduction in the level of either protein would explain our results.

- Line 210 – Is there an obvious reason why less SurA is required compared to previously reported in vitro folding assays? Could this be explained by access (e.g., the OMV-based assay necessitated getting SurA inside the vesicles)?

For the sake of clarity, we have now modified the text (see lines 215-218) to indicate that the 8:1 ratio of SurA to BAM is similar to the ratio that we used (and that worked well) in previous experiments in which we analyzed the assembly of OMPs into BAM proteoliposomes. We do not know why the OMV-based assay requires so much SurA.

- Line 217 and Figure 2c – The discussion notes that how the substrate is produced can explain the inability of Skp and/or DegP to facilitate folding of EspPΔ5'. Could the role of these chaperones also be substrate dependent (e.g., could they play a role with different OMPs)? More generally, have you measured the folding of trimeric OMPs with your in vitro assay?

That's an interesting question. So far we have only examined the ability of different chaperones to facilitate the assembly of EspPΔ5', and it is indeed possible that the role of chaperones is substrate dependent. We showed that SurA is required for the assembly of OmpC into BAM proteoliposomes (see ref. 66), but we did not test any other chaperones.

- Line 237 – The observation that BAM density can dictate activity is intriguing. Did you test whether this deficit in BAM activity in the uninduced sample be overcome by increasing the amount of substrate and/or SurA?

So far we have not tested that possibility, but it would be interesting to do so in future studies. In previous experiments we have found that increasing the SurA to substrate ratio beyond 10 does not significantly increase assembly efficiency.

- Figure 2c – It looks like there is less overall EspPΔ5' in some lanes (e.g., DegP, SurA+Skp, SurA+DegP). Is there an explanation for this (e.g., degradation)?

To address the reviewer's concern, we have replaced the results shown in Fig. 2C with the results of a different replicate of the same experiment in which the levels of EspPΔ5' are similar in every lane. The reduced level of EspPΔ5' in some of the lanes in the original version of the figure appears to be a quirk that resulted from the use of a different lot of the PURExpress kit.

- Line 273 – More of an observation than a question, but you point out that the t_{1/2} for assembly are slower for TN102, but is there any reason to think that the ~3.5-minute t_{1/2} versus the ~2.5-minute t_{1/2}

for the other three strains is biologically meaningful? Perhaps not given that Figure S8 indicates that the OMP levels (except for, perhaps, FepA) are more or less the same (at least under the tested growth conditions).

We mulled over that question ourselves, and we really do not know if a ~1.4-fold difference is biologically significant. Nevertheless, we show in Fig. 4C that the p value is <0.01. Furthermore, we wrote the section that begins with line 282 cautiously and state that the results “suggest” that the OMs purified from TN102 affect the functionality of BAM. They certainly do not prove anything.

- Line 399 and Figure 6 – You note that the effects of lacking MlaA and PldA are synergistic. Is this the right description? TN103, I think, looks like a distinct pattern from each of the individual mutants, but perhaps I am misinterpreting this?

The reviewer is correct in noting that the deletion of both *miaA* and *pldA* does not have a synergistic effect across all classes of lipids. The deletion of both genes has a synergistic effect on the levels of long chain PGs, PEs, and CLs (in both OM and WC samples), but not on the levels of short chain PLs (for which the lipid profile of TN103 more closely resembles that of TN101 across all assayed lipid classes) or lysophospholipids (for which the TN103 pattern matches the pattern of TN102). We have now modified the text on lines 412-414 and 421-423 to clarify this point.

- A more general question: did you run lipidomic analysis of outer membranes isolated using the sucrose gradient approach? The reported comparisons with the whole cell preparations are extremely helpful, but one point that remains is whether the sarkosyl detergent has any added effect on the isolated lipid composition. Assessing the phospholipid composition in a detergent-free preparation could be informative for this question.

Although we agree with the reviewer that in principle it would be interesting to perform a lipidomic analysis of OMs isolated in the absence of sarkosyl detergent, we were concerned that the synthesis of EspPΔ5' (a model OMP) was somewhat variable in the presence of OMs purified by sucrose gradient fractionation (see lines 199-203). We suspect that there are unknown contaminants in the OMs (possibly including some inner membrane vesicles) that interfere with our *in vitro* transcription/translation reactions that might also complicate the interpretation of the results of lipidomic experiments.

- A minor suggestion to help with readability throughout - The authors could consider adding the relevant genotypes whenever the strains are noted in the text (for example: NR698 lptD4213, TN101 Δ*miaA*, TN102 Δ*pldA*, TN103 Δ*miaA* Δ*pldA*) so the reader does not have to continually refer back to the original strain descriptions.

We have added relevant genotypes throughout the text and Figures to improve readability as suggested.

- Again, more of a comment than a question, but I am very interested to see future applications of this OMP folding assay and lipidomic measurements performed under more perturbative conditions that might reveal additional effects of PldA and MlaA and insight into the mechanism of BAM.

We agree that it would be interesting to use our assay to obtain further insight into the importance of PldA and MlaA under stressful conditions. To emphasize this point we have added the words “or stress” to line 554, which now reads “...our results might reveal the effect of significant changes in the OM PL

profile on OMP assembly that occur *in vivo* under specific growth or stress conditions...”

Reviewer #3 (Remarks to the Author):

This manuscript by Nilaweera et al present data that raises some interesting questions about the role of phospholipids in the outer membrane (OM). Following optimisation of conditions for Bam expression in OM fractions they analyse activity as judged by OMP folding. This contrasts previously used methods of proteoliposomes, providing an alternative method for functional analysis of Bam. Similar to previously reported data results showed SurA is essential for folding of EspP (primary read out of activity in manuscript). Nilaweera and team observe that induction of Bam during production of OM fractions is more active than the same concentration of Bam from a uninduced sample. This aligns with other data in the literature that shows OMP clusters to be important for membrane stability and organisation. I found the latter sections to become more noteworthy as Bam activity was compared between mutant strains with impaired lipid transport. PldA deletions had the most pronounced effect on Bam activity reducing folding from 37% in WT to 7%. MlaA and LptD deletions had a weaker effect reducing the rate of activity without impacting amount of protein folded as significantly. Elegant LC-MS/MS experiments were developed to study the proteoliposomes of the mutants. This showed that PldA mutants had an OM that resembled the IM PL profile, whereas Mla was similar to WT cells. This supports the previous idea that each system has distinct roles in the mechanism for removal of PL from the OM.

I found the experimental procedures to be largely robust, and the material, particularly the LC-MS/MS, was both novel and interesting. The data clearly demonstrate that phospholipids influence Bam activity within this *in vitro* system. The authors appropriately acknowledge that further investigation is needed to elucidate the *in vivo* relevance of these findings. It remains to be determined whether the phospholipids themselves are directly involved in protein interactions, or if the observed effects are due to lipid-induced changes in membrane fluidity and disruption of outer membrane organisation and stability. I have outlined several points below where further clarification would enhance the manuscript.

We thank the reviewer for his/her positive and insightful comments.

Further discussion and clarification required:

- Line 238-239 and 435-442: Native levels of Bam provide sufficient activity to fold OMPs *in vivo*. Could the phenomenon that is observed here be related to observations in this paper: <https://www.pnas.org/doi/10.1073/pnas.2414725122> That mentioned OMPs are largely mobile so to create the clusters Bam has to be concentrated in these precincts to maintain OM organisation and lipids can insert anywhere.

The main conclusion of the paper mentioned by the reviewer is that OMP insertion is mid-cell biased while LPS localization is much less restricted. The authors also corroborate an earlier finding that BAM forms clusters that are randomly distributed throughout the OM. Based on results described in ref. 42, the formation of these clusters appears to be associated with BAM activity. As we note on lines 448-450, BAM might be more active in our system when it is overproduced because higher levels of the complex form clusters more readily. We do not believe that the role of BAM in maintaining OM organization, however, is directly related to our study. Nevertheless, to address the reviewer’s comment we now state that BAM forms clusters (that might be associated with the formation of “OMP islands”) in the Introduction (lines 89-90) and cite the paper mentioned above.

- Line 464-466: could the authors comment on inclusion or exclusion of the possibility that the specific PL enrichment causes a more fluid membrane where the organisation of the proteins is not maintained. It is this organisation that impacts activity rather than the PL themselves. Neighbouring OMPs creates a more rigid membrane that aid folding rather than lipids, particularly phospholipids that will create a very fluid environment, like the IM, and lower insertion rates due to a lack of supporting structure around Bam.

We believe that changes in the PL profile might influence the properties of the OM (or any biological membrane for that manner) in many different ways. As the reviewer suggests, specific PL enrichment/depletion might alter OM fluidity and/or OMP organization, but to mention a few other possibilities, it might also influence membrane curvature, the formation or size of LPS islands, OMP mobility, OMP structure, or OMP function. Some or all of these changes might affect BAM activity. In lines 464-466 (now lines 478-482) we focus on the enrichment of lyso-PLs only because previous studies have presented evidence that these lipids affect membrane remodeling, which is highly relevant to BAM activity. We believe that our collective understanding of the functions of other PLs is too limited to speculate on the ways that they might affect OMP assembly. We should note that the sentence that precedes lines 478-482 (“our results emphasize that PLs...can shape the properties of each membrane and/or serve distinct roles in cell physiology”) was deliberately written to be vague because we do not have space to consider a wide range of possibilities.

- Line 415-416: Could the authors clarify why membrane crosstalk is concluded as the primary explanation, rather than the alternative that Mla mutations have a weaker effect because phospholipids from the inner membrane are efficiently removed by the Pld? When the Pld is absent, Mla that might remove phospholipids with specific acyl chain lengths cannot fully compensate, leading the OM to adopt a phospholipid profile resembling that of the IM. Rather than crosstalk, this may reflect distinct, complementary roles of the two phospholipid removal systems, each influencing the lipid profile in different ways.

Consistent with the reviewer’s comment, we note in the Discussion (lines 507-509) that the *pldA* mutation has a more significant impact both on the PL profile and BAM activity than the *miaA* mutation. Our goal at this point in the text (and at the end of the Results) is to explain the synergistic effect of the *pldA* and *miaA* mutations, that is, the observation that the combination of the two mutations causes a more significant change in the OM PL profile than either mutation alone. The problem is that the loss of both PldA and MiaA is not simply the sum of the loss of PldA alone plus the loss of MiaA alone. As shown in Fig. 6B, the change in the level of each lipid in the absence of MiaA (column 3) and the change in the level of the same lipid in the absence of PldA (column 4) do not add up to the change observed in the absence of both MiaA and PldA (column 5). Something more complicated must be going on that we believe could be due to an unknown regulatory mechanism or even crosstalk between the IM or OM. To focus on this possibility and to address the reviewer’s concern, we have now added the words “but non-additive” to the beginning of the sentence on lines 427-431, which now reads “The synergistic (but non-additive) effect of the two mutations...”

Minor points:

- A general point: the strain names are hard to remember what the mutation is, using a short hand such as Δ MiaA would make it easier for the reader.

As suggested, we have now inserted the strain genotypes next to the strain names in several places in the text and in the Figure labels to improve the readability of the manuscript.

- Line 37: It is well established that MlaA and PldA have distinct roles in phospholipid removal from the OM. It may be helpful to revise the wording to reflect their non-redundant functions more clearly.

Most of the evidence (at least that we are aware of) that MlaA and PldA play distinct roles in PL removal was published in ref. 43 (Malinverni and Silhavy, PNAS 2009). In that study the authors showed that *mLaA* and *pldA* knockout strains have different phenotypes in genetic experiments. Because the genetic results might be considered to be “indirect evidence” while the results of our OMP assembly assays and lipidomic analysis could be considered to be “direct evidence” in support of the idea that MlaA and PldA have distinct roles in PL removal, we changed the word “clear” to “direct” on line 37. We believe that this modification will help to distinguish our work from previous work.

- Line 54-56: Whilst OMPs are not conventional alpha helical membrane proteins, their conservation and high copy numbers in these bacteria would suggest that they are perhaps not that “unusual”. Revised wording could be beneficial.

Good point. In response to this comment we have changed “unusual” to “distinct”.

- Line 89-90: Please check the wording here as it is unclear. Suggest removing the “however”.

As suggested, we have removed the word “however”.

- Line 92: While I appreciate the authors' emphasis on the importance of membrane asymmetry and acknowledge that certain OMPs can be deleted without completely inhibiting growth, there are important caveats. For instance, components such as Bam and Lpt are essential, and cells lacking core OMPs exhibit significant stress or impaired viability. I suggest rephrasing the sentence to reflect these nuances.

As suggested, we changed the sentence to “Only a few OMPs are essential for viability or OM integrity...” (line 94). Only BamA and LptD are essential, and the core OMPs that the reviewer refers to (e.g., OmpA) fit into the latter category.

- Line 121-122: OMVs should have a fully intact OM including asymmetry and OMP profile representative of the region of the cell from which it budded off.

We agree with the reviewer, but as shown in previous studies that are cited on line 131, OMVs do not contain an OMP profile that is representative of the entire cell surface. For clarification, we changed the word “distribution” on line 121 to “sampling” and inserted “full complement of” before “resident OMPs” on line 125. See also our response to the next comment.

- Line 130: The authors state that outer membrane (OM) fractions provide a more representative view of the OM landscape than outer membrane vesicles (OMVs). However, the images of OM fractions prepared using Sarkosyl show the formation of small, vesicle-like structures. Given that OMVs retain intact membranes, one could argue that they share similar structural features. The challenge with vesicles lies in the uncertainty regarding their origin, are they derived from specific regions of the cell envelope and do they contain outer and/or inner membrane components. A potentially stronger and

more defensible point would be that the OM fraction may offer broader sampling of the OM across the entire cell surface, thereby providing a more comprehensive overview.

We most certainly agree with this statement. To incorporate the reviewer's thinking into our manuscript, we have now modified lines 134-135 to indicate that in our study we used "purified OM fractions that provide a more comprehensive representation of the native OM landscape than BAM proteoliposomes, OMVs or EMM..."

- Line 131 vs 135: OM fractions referred to native OMs got confusing on the way through. Further clarification, native OMs is referring to the intact nature through use of Sarkosyl not native meaning wild-type vs mutant.

For clarification we have slightly modified the text. We now point out that the term "native OMs" is simply our own designation and is not a standard term (line 136). We would have liked to use the name "outer membrane vesicles", but that name has already been taken!

- Line 174: Will the observed effect be due to overall OM functionality, or specifically a result of Bam complex activity? Please rephrase to clarify this distinction.

We started with the assumption that OMP assembly would be due to the activity of BAM in the purified OM and subsequently validated this assumption by showing that the OMP assembly that we observed is blocked by the BamA inhibitor darobactin. We have now clarified the text (lines 179-180) by stating that our goal was to "evaluate the functionality of the BAM in the purified OM samples".

- Line 184/601-602: Much of the analysis and comparison presented relies on accurately determining the Bam concentration within the samples. While western blotting is one of the few feasible methods for estimating protein levels in a complex mixture like the OM fraction, it is often limited by variability and the quality of the antibodies used. It would be helpful if the authors could include a control blot in the supplementary material, showing OM fractions with Bam at a few specified concentrations across samples (e.g., induced vs. non-induced conditions). This would help demonstrate the reliability of the measurements since 1–2 μM Bam is relatively low, and showing this would support the observed differences in downstream folding assays.

To address the reviewer's concern, we have added a new Supplementary Figure (Fig. S3) to demonstrate the reliability of our measurements of the concentration of BAM in native OMs. We ran several concentrations of purified BAM on a gel (where the concentrations were determined at A_{280} using the published value of $\epsilon = 294,630 \text{ M}^{-1} \text{ cm}^{-1}$) and then used the intensity of the signals observed on a Western blot performed with anti-BamA to generate a standard curve. In parallel, we ran three different amounts of native OMs obtained from MC4100 in which BAM expression was induced or not induced on a second gel, performed Western blots with anti-BamA, and determined the concentration of BAM using the standard curve. In essence, the results show that "low" concentrations of BAM can be accurately determined using our Western blotting method. As an aside, 1-2 μM BAM is higher than the concentration that we have used in previous studies. We have typically performed assays with proteoliposomes that contain 0.5 μM BAM and observed a similar level of EspP $\Delta 5'$ assembly (~40-50%).

- Line 186: How many lysine residues are modified per EspP molecule, and is the extent of modification consistent across all molecules? I ask this to assess whether the modification might influence folding

kinetics. If the degree of BODIPY labeling on lysines is uniform across samples, that would help support the validity of the folding comparisons. Was this assessed or controlled for?

There are 20 lysine residues in EspP Δ 5'. Because we added only a small amount of BODIPY labeled lysine to our assays and there was a large excess of unlabeled lysine, only a small number of lysines (perhaps ~0-2) were labeled in each molecule. As we state in the text (line 190), the labeled lysine residues should have been incorporated into the protein randomly. It seems likely that some of the EspP Δ 5' molecules contained a modified lysine that affected their rate of folding, and some molecules might not have folded at all. In our kinetic analysis (Fig. 4) the only difference was the source of the native OMs. Because the distribution of modified forms of EspP Δ 5' that were synthesized should have been essentially identical in each reaction, the effect of specific modifications on the folding of the protein does not impact our interpretation of the results.

- Figure 2: I generally really like the figures in this manuscript. I just wonder if 2a and 2b might be better placed in the supplementary material?

We believe that Fig. 2A and 2B will help readers who might not be familiar with our work understand our results, and for that reason should remain in the main manuscript. Nevertheless, in thinking about the reviewer's comment, we realized that it would be helpful to show the site of the autocatalytic cleavage in EspP Δ 5' in Fig. 2A.

- Figure S3: Could the authors add to the figure caption to define what the control sample is?

The control sample contains no OMs. For clarification we have now changed "control" to "no OMs" in the labeling of Fig. S3A (now S4A) and S1D.

- Line 210: Looking at the figure it could be argued that anywhere from 4-10 μ M is optimal, could the authors comment on how significant the difference is between these points to clarify why 8 μ M was defined as optimal

We used 8 μ M SurA, but we believe that it would be fair to say that based on the results shown in Fig. S5 both 8 μ M SurA and 10 μ M SurA promoted maximum assembly. We have removed the word "optimal" and modified the text to reflect the results more accurately (see lines 215-218).

- Line 258: How much peptidoglycan would be expected to remain in the preparation? If present, it's worth noting that mature peptidoglycan has been shown not to significantly affect activity. Therefore, the observation that lysozyme treatment has little impact may not be unexpected. Clarifying this point could help contextualise the result.

We really do not know how much—if any—peptidoglycan remained in our sample preparations. We performed the experiment shown in Fig. S7 to rule out an alternative explanation of our results, even if that explanation might be unlikely. To soften our statement (but to keep it concise) we changed "residual peptidoglycan" to "any residual peptidoglycan" (line 265).

- Line 265: NR698 "appears" to fold more EspP than TN101 but less than wild-type in figure 4. To me this suggests that the MlaA mutation has a more pronounced effect than the LptD mutation. Combined with the time course in figure S7 NR698 looks more comparable to WT whilst MlaA lags until time point 30min. Similarly in Fig 4c TN101 looks to lag more after that 50% maximum assembly point. Are the

approximate time differences and % of EspP folded significant and if so could the authors clarify on how this was calculated/concluded. It will help to recapitulate what is observed at first glance of the figures with the text.

We agree that there are some noticeable differences in the assembly of Esp Δ 5' in the presence of OMs purified from NR698 and TN101, but we do not think that we would be justified in claiming that there is a significant difference in the levels of BAM activity. After all, the p values for the kinetics of assembly in the presence of OMs purified from NR698 are rather high, and we cannot be certain that the modest differences in the two curves are not simply due to chance.

- Line 275-276: TN103 appears to show impaired Bam function similar to TN102, could the authors clarify this within the text here?

In this sentence we are only stating that the difference in the $t_{1/2}$ of assembly in the presence of OMs purified from TN102 and OMs purified from wild-type cells was particularly notable. We state that we observed a similar impairment of BAM function when we purified OMs from either TN102 or TN103 earlier in the text (see lines 263-264).

- Fig S9/Line 290: Could the authors speculate why LPS-deficient strain NR698 produce smaller vesicles compared to the controls? Understanding the underlying cause would provide valuable insight into membrane dynamics in these mutants.

As suggested by the reviewer, we now speculate on the reason that the NR698 vesicles were smaller (lines 302-303). NR698 colonies have an unusual morphology, and the cells are hypersensitive to antibiotics. These properties might be connected to the small vesicle size.

- Fig S10 and Fig S8. There appears to be some variability in the levels of outer membrane proteins (OMPs) such as BtuB, FepA, LamB, FadL, and OmpT in the purified OM samples relative to whole cell. Could the authors comment on the repeated data in WC samples and why S8 was not done with OM fractions that were analysed throughout? Specifically, Line 284–286 states that OMP levels are consistent, which seems to hold true in WC samples, but may not fully apply to purified OM fractions. Given that mutations leading to increased phospholipid content likely impair the OM and promote OMP shedding during detergent treatment, it is surprising that TN102 with pronounced effects on Bam function appears to have higher OMP levels in OM samples than TN101.

The reviewer makes a good point here. Although Fig. S8 (now Fig. S9) shows that the levels of all of the OMPs that we tested are similar in all of the strains—and the three replicates of the same experiment that we performed in response to a concern raised by reviewer 1 bolster our original observation—we did not analyze the protein content of the purified OM fractions until Fig. S10 (now Fig. S11). Our primary goal in Fig. S8 was to make an initial assessment of the effects of the mutations on OMP levels and to confirm the results that had been previously reported. To both address the reviewer's comment and to ensure that the samples that we used for our lipidomics analysis differ primarily in their phospholipid profiles, we normalized the protein levels in the mutant strains in all four of the replicates we generated for our lipidomics analysis to the levels observed in MC4100 and then performed a statistical analysis (see Fig. S11C-D). There is a bit more variability in the protein levels in the purified OMs than in the WC samples, but that is not surprising because the purified OMs undergo more processing than the WC samples and are therefore more prone to experimental errors. The key points here are that 1) the average levels of the most abundant proteins (OmpC and OmpA) in all the mutant

strains are within ~20% of the average levels of MC4100 and 2) while the average levels of the other proteins (which are much less abundant) differ from their average levels in MC4100 by ~20-~50%, there is no correlation between the differential and either the activity of the native OMs in OMP assembly assays or the degree to which the OM phospholipidome is remodeled. For example, the level of BamA in the OMs purified from NR698 and TN101 deviated the most from the level in MC4100, but the mutations in those strains produced the smallest effects on OMP assembly and the composition of the phospholipidome. Likewise, the levels of FadL in the OMs purified from all of the strains clustered together except TN102, but based on the observed effects of the mutations on OMP assembly and the composition of the phospholipidome the *pldA* knockout in TN102 should have clustered with the *pldA mlaA* double knockout in TN103. In any case, to describe the results more accurately we have modified the sentence that begins on line 357 to now read “In a control experiment we found that the levels of two highly abundant OMPs (OmpC and OmpA) were similar in all of the samples we used for our lipidomic analysis while there was slightly more variability in the levels of less abundant OMPs in the OM samples that appear to be due primarily to experimental error associated with sample preparation (Fig. S11)”.

As an aside, we now note that in our lipidomics analysis the signal of each phospholipid was normalized to the sum of the phospholipid signals in that sample (lines 327-328), which is equivalent to normalizing each phospholipid signal to the phospholipidome itself. This strategy was taken to offset sources of variation in sample preparation that might have inserted artifacts into the statistical comparisons.

- Line 294: It is not convincing at the resolution shown in images that there is a “double-layer” to vesicles. Higher magnification and resolution would be required to make this convincing.

To address this concern we have toned down the text, which now reads “appeared to have” a double-layer morphology (line 304). As it turns out, the TEM studies were performed at the highest level of magnification that the available microscope can achieve.

- Line 348-349: can the authors comment on why LPS is consistent across all strains even in the LptD insertion machinery mutant?

We agree with the reviewer that it is curious that the defect in LptD does not lead to a reduction in the level of LPS, but our results are consistent with those of other investigators, some of which were reported in ref. 89, Fig. S3, and some of which were described to us as a “personal communication” by the lead author of that paper. She also told us that there is a complex regulatory mechanism that determines the level of LPS that explains why many defects in the Lpt system do not affect the level of LPS. We believe that this topic is beyond the scope of our study, especially because we only wanted to determine if a change in the level of LPS might account for the differences in OMP assembly activity that we observed. With respect to the other strains, there is no obvious reason to suspect that mutations in *mlaA* or *pldA* would affect the level of LPS.

- Line 447: could authors expand on how number of active BAM complexes is discerned from consistent number of Bam complexes with reduced rate?

To address the reviewer’s concern, we modified the part of the Results section in which we first discussed the functionality of the BAM that resides in the OMs purified from the mutant strains. In this section we first state that the rate of assembly of EspPΔ5’ in the presence of OMs purified from TN102 was notably slower than the rate of assembly in the presence of wild-type OMs (lines 281-282). We then

state (line 282-284) that the “kinetics data” (changed from “these results”) “suggest that changes in the native OMs purified from TN102 affect the functionality of BAM (i.e., its rate of catalysis).” We added the words in parentheses to help explain what we mean by “BAM functionality”. We then contrast OMs purified from TN102 with those purified from the other mutant strains, which promote EspPΔ5' assembly at a rate that is closer to the rate promoted by wild-type OMs. Because the rates are similar, we think that it will be apparent to readers that the significant reduction in the levels of EspPΔ5' assembly that we observed are due more to a reduction in the number of active BAM complexes than a change in the rate of catalysis.

As an aside, we have tried to write lines 281-288 very carefully, especially because the rate differences are not that sizable. Our goal is simply to point out some potentially interesting nuances in our data.